# Quantitative fragmentomics allow affinity mapping of interactomes

Gergo Gogl [1] ✉, Boglarka Zambo [2], Camille Kostmann[1],
Alexandra Cousido-Siah[1], Bastien Morlet [2], Fabien Durbesson[3], Luc Negroni [2],
Pascal Eberling[2], Pau Jané[1], Yves Nominé [1], Andras Zeke [4],
Søren Østergaard [5], Élodie Monsellier[1], Renaud Vincentelli[3] & Gilles Travé [1] ✉

Human protein networks have been widely explored but most binding affinities remain unknown, hindering quantitative interactome-function studies. Yet interactomes rely on minimal interacting fragments displaying quantifiable affinities. Here, we measure the affinities of 65,000 interactions involving PDZ domains and their target PDZ-binding motifs (PBM) within a human interactome region particularly relevant for viral infection and cancer. We calculate interactomic distances, identify hot spots for viral interference, generate binding profiles and specificity logos, and explain selected cases by crystallographic studies. Mass spectrometry experiments on cell extracts and literature surveys show that quantitative fragmentomics effectively complements protein interactomics by providing affinities and completeness of coverage, putting a full human interactome affinity survey within reach. Finally, we show that interactome hijacking by the viral PBM of human papillomavirus E6 oncoprotein substantially impacts the host cell proteome beyond immediate E6 binders, illustrating the complex system-wide relationship between interactome and function.

Over the past decade, proteome-wide mammalian interactomics have identified few hundreds of thousands of binary interactions[1–4]. However data are mostly qualitative, lacking quantitative information on binding strengths, and the coverage of the human interactome remains incomplete. Remarkably, protein interactomes are built upon minimal interacting blocks, consisting of globular folded domains and short disordered linear motifs[5–9]. These fragmental interactions occur both intermolecularly and intramolecularly[5]; their intrinsic affinities, fine-tuned by evolution, rule cooperation, competition and specificities underlying most cellular functions[10,11]. They are also key targets for pathologies such as microbial infections and cancers[12–15]. The accurate description and modeling of complex biological systems and

their pathological defects will ultimately require quantitative affinity measurements of fragmental interactomes at proteome-wide scale.

The PDZ-PBM interactome is a relevant model for such studies. The human proteome comprises 266 human PDZ (PSD95/DLG1/ZO1) domains [16] and about 4000 putative PBMs, defining a network of one million potential interactions. PBMs are mostly (though not exclusively) C-terminal, with their COO⁻ implicated in binding, and are classified based on position −2, being respectively Ser/Thr, hydrophobic or acidic in classes 1, 2, and 3 [17,18]. PDZ-PBM interactions are rather transient and promiscuous[19–22]. The PDZ-PBM interactome is also prone to pathological perturbations such as viral infection and cancer[13]. Viral proteins bearing functional PBMs are found notably in

[1]Équipe Labellisée Ligue 2015, Département de Biologie Structurale Intégrative, Institut de Génétique et de Biologie Moléculaire et Cellulaire (IGBMC), INSERM U1258/CNRS UMR 7104/Université de Strasbourg, 1 rue Laurent Fries BP 10142, F-67404 Illkirch, France. [2]Institut de Génétique et de Biologie Moléculaire et Cellulaire (IGBMC), INSERM U1258/CNRS UMR 7104/Universite de Strasbourg, 1 rue Laurent Fries BP 10142, F-67404 Illkirch, France. [3]Architecture et Fonction des Macromolécules Biologiques (AFMB), UMR 7257 CNRS-Aix-Marseille Université, Marseille, France. [4]Bioinformatics Research Group, Research Centre for Natural Sciences, Magyar tudosok korutja 2, 1117 Budapest, Hungary. [5]Novo Nordisk A/S, Global Research Technologies, Novo Nordisk Research Park, 2760 Maaloev, Denmark. ✉e-mail: goglg@igbmc.fr; traveg@igbmc.fr

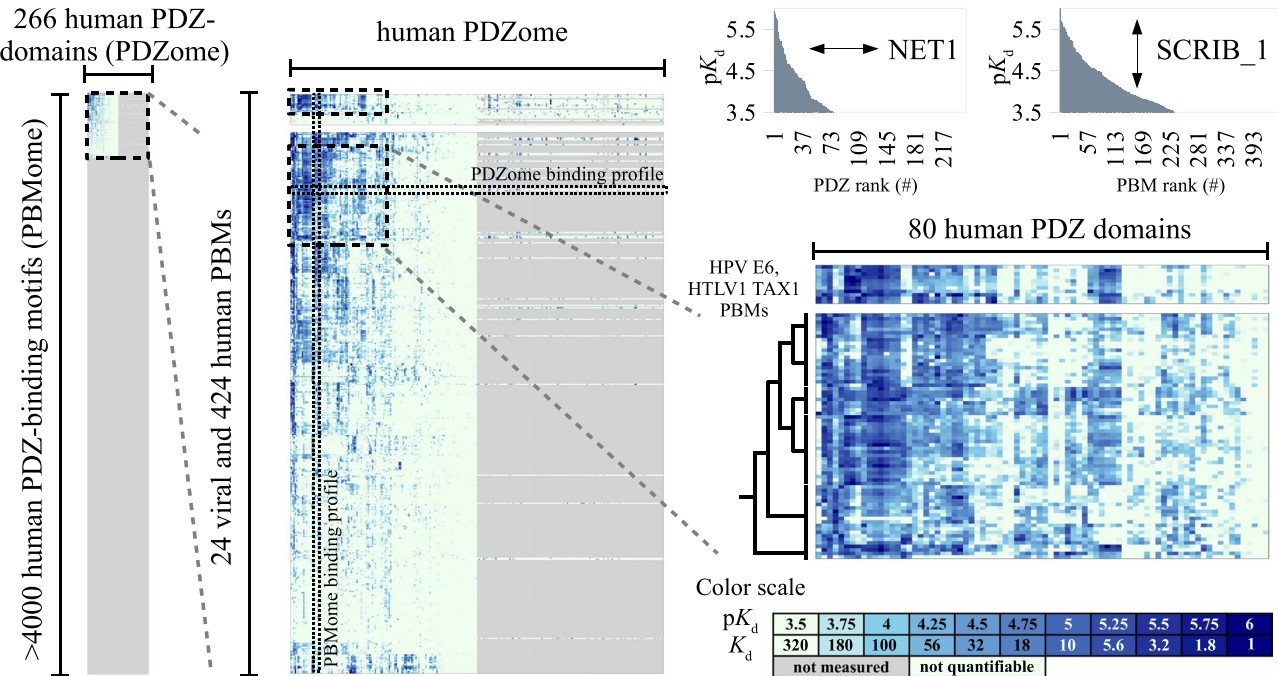

**Fig. 1 | Affinity profiling of a subsection of the PDZ-PBM interactome: principle and data overview.** The human PDZ-PBM interactome represents ~10⁶ potential interactions (left panel). Focusing on a viral- and cancer-relevant region (upper part of left panel, zoomed in middle panel), we measured the 59,578 affinities of 133 human PDZs for 424 human and 24 viral PBMs (fully measured subsection, left half of middle panel) and 5,573 extra affinities involving the remaining 133 PDZs and a subset of 45 PBMs (right half of middle panel). The figure does not indicate the scale of all potential viral hijacking interactions. Measured or non-measured PDZ-PBM pairs are respectively colored in heat map mode (see color scale in lower right panel) or in gray. Any horizontal or vertical cross-section of that interactome represents an individual PDZome- or PBMome- binding profile, as illustrated for NET1 PBM and SCRIB_1 PDZ (top right corner). The horizontal cross-section of the middle panel is indicated by a horizontal arrow in the corner of the PDZome binding profile and the vertical cross-section of the middle panel is indicated by a vertical arrow in the corner of the PBMome binding profile. Affinity-based Euclidian distances computed from the fully measured subsection reveal a clade of human PBMs displaying the highest inter-actomic similarities with HTLV1 Tax1 and HPV E6 oncoviral PBMs (middle right panel). Note the general similarity of the heatmap patterns of the oncoviral PBMs and of the identified human PBM clade, and the fine pattern differences further revealed by sub-clustering using the UGPMA approach. For more details, see Supplementary Data 1.

HPV and HTLV oncoviruses as well as in HBV, WNV and coronaviruses including MERS or SARS-CoV2[13,15,23–26].

Here we measured with unprecedented coverage and sensitivity ~65,000 PDZ-PBM affinities in a defined region of the human inter-actome targeted by viral PBMs, including those of oncogenic proteins HPV E6 and HTLV1 Tax1. The data, assembled into an open-access database (https://profaff.igbmc.science), define a quantitative inter-actomic space where proteins are located and clustered according to their binding preferences. Using these data, PDZ-PBM specificities can be depicted by binding profiles and specificity logos that we interpret by crystallographic studies on selected instances. We also show that amounts of prey proteins captured in cell lysates by exogenous PDZ or PBM baits correlate with the corresponding PDZ-PBM affinities. Finally, we investigated how the PDZ interactions of the viral PBM-containing HPV E6 oncoprotein impact the whole host cell proteome.

## Results

### Large-scale affinity mapping of the PDZ-PBM interactome

To explore the PDZ-PBM interactome, we expressed a recombinant PDZome library covering all the 266 known human PDZs[27], and we synthetized a 10-mer peptide library of 24 viral and 424 human PBMs, of which 323, 63, 51, and 11 belong to class 1, 2, 3, and to atypical or non-C-terminal subgroups, respectively (Supplementary Data 1). Eight PBMs harbored post-translational modifications (phosphorylation or acetylation) which may modulate binding specificities[20–22]. The viral PBMs included notably 12 PBMs from oncoproteins HPV E6 (11 distinct types) and HTLV1 Tax1. The host PBM list was designed to explore the closest interactomic neighborhood of these oncoviral PBMs, while attempting to sparsely yet evenly cover the sequence diversity of human PBMs.

To quantify dissociation constants, we used the holdup method, a high-throughput comparative chromatographic retention assay that we developed previously[19,22,28]. The assay measures the total and unbound concentrations of reactants at equilibrium, which can be converted into steady-state dissociation constants herein reported as $pK_d$ values (the negative of the base 10 logarithm of the dissociation constant). To adapt the method for higher throughput, we implemented several technical improvements and rigorous benchmarks into our previous protocol (see Methods). The current protocol allows measuring up to 10,000 distinct domain-motif pairs per day, and quantifies their affinities at remarkable sensitivity. $pK_d$ quantification thresholds, defined as the limit above which affinity constants could be quantified in each assay, were mostly comprised between 4 ($K_d = 100$ μM) and 3.1 ($K_d = 800$ μM). In total, we performed 79,374 single-point holdup experiments on 65,151 interactions, covering ~55% of the interactomic space defined by the 266 human PDZ-domains and the 448 human and viral PBMs (Fig. 1). We particularly focused on 133 PDZ domains representing the strongest partners of the HPV E6 and HTLV1 Tax1 PBMs. We quantified 18,332 unique dissociation constants, whereas 46,825 PDZ-PBM affinities representing 72% of the explored space remained below the assay's quantification threshold.

The complete affinity data (Supplementary Data 1) are freely accessible on our online database ProfAff (for "Profiling Affinities") (https://profaff.igbmc.science), allowing user-friendly visualization and analyses.

We complemented our measurements with 395 detailed competitive fluorescence polarization experiments (FP). The quantified affinities show an excellent agreement with those measured by holdup assay (Fig. 2A). We also assembled, expanding upon a recent review[29], a literature compendium of affinities previously measured for

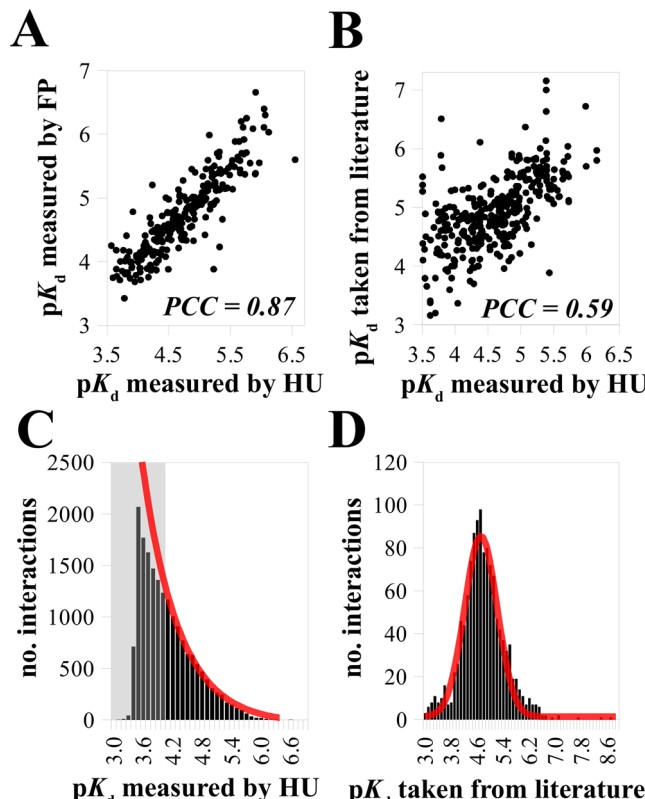

**Fig. 2 | Validation and distribution of the quantified PDZ-PBM affinities.**
**A** Comparison of PDZ-PBM dissociation constants quantified with holdup assay (HU) or competitive fluorescence polarization (FP). The Pearson correlation coefficient (PCC) was determined based on affinities quantifiable by both methods (*n* = 255). **B** Comparison of PDZ-PBM dissociation constants quantified with holdup assay or obtained from literature. PCC was determined based on interactions quantified by both methods (*n* = 362). **C** PDZ-PBM dissociation constants quantified from HU follow an exponential-like distribution. The gray zone indicates the range of p$K_d$ quantification thresholds that slightly varies between different assays. The red line indicates an exponential probability distribution function, predicted from the data. **D** PDZ-PBM dissociation constants obtained from literature follow a normal distribution. The red line indicates the predicted normal distribution. Note the different affinity range in **C** and **D**. Source data are provided as a Source Data file. For more details, see Supplementary Data 1.

mammalian PDZ-PBM pairs (Supplementary Data 1). Considering unavoidable experimental discrepancies (methods, exact lengths of constructs and peptides, species of origin…) our dataset remarkably agrees with this compendium (Fig. 2B).

Empirically, we have found that the quantified PDZ-PBM affinities follow a nearly perfect exponential-like distribution (Fig. 2C). A handful (<0.5%) show p$K_d$ > 6, only 8% have a p$K_d$ > 5, and the majority has a 3 < p$K_d$ < 4. Remarkably, the affinity histogram starts to diverge from the exponential trend at p$K_d$ < 4, precisely when entering the gray zone where some affinities start to be below quantification threshold (Fig. 2C). By contrast, the literature benchmark is greatly underrepresented in weak affinities and follows a normal distribution centered around 4.7 p$K_d$ with a width of 1 p$K_d$ (Fig. 2D).

We curated and assembled from the proteome-wide interactomic resource BioPlex[3] and the BioGRID and IntAct interaction databases[30,31] a second literature benchmark based on qualitative interactions devoid of affinity data, involving full-length PDZ- and PBM-containing proteins or their fragments (1651 interactions, see Supplementary Data 1 and Supplementary Fig 1). This qualitative protein interactome benchmark comprised 1248 interactions overlapping with our quantitative PDZ-PBM dataset, and it included 725 interactions (58%) that could be assigned to at least one quantified fragmentomic affinity.

We also found that only 0.5%, 2.1%, and 3.4% out of the 14,839 protein pairs for which we quantified PDZ-PBM affinity constants corresponded to protein pairs recorded by BioPlex[3], IntAct[31], and Biogrid[30], respectively. Our exhaustive affinity survey of a defined interactomic space thus indicates that the current coverage of the human interactome remains very sparse, at least for interactions involving short linear motifs.

## Topology of the human PDZ-PBM interactomic space

p$K_d$ values can be considered as coordinates of multidimensional interactomic spaces. Our dataset almost exhaustively covers two such spaces, one describing PDZ-binding by PBMs and the other describing PBM-binding by PDZs, respectively comprising 133 and 448 affinity dimensions. The more two PBMs or two PDZs bind to the same targets with similar affinities, the closer they stand in these spaces. Euclidian distances between PDZs or between PBMs can be calculated from their differences in affinities ($\Delta$p$K_d$ = $\Delta\Delta G$/2.303RT, see Methods). These distances allow identifying closest interactomic neighbors (Supplementary Fig 2A). For example, the PDZ-binding profiles of PBMs of the RhoA guanine nucleotide exchange factor NET1 and of the Ras effector protein RASSF6 are extremely similar, and this is captured by a very short Euclidian distance. Human PBMs that are closest interactomic neighbors of viral PBMs are of particular interest, as they represent their most potent rivals to bind the same host PDZs with similar affinity properties. For instance, the closest neighbors of the major oncoproteins HPV16 E6, HPV18 E6, and HTLV1 Tax1 in our explored interactome are CYSLTR2, NET1, and GRIN2C, respectively.

We clustered PBMs based on their Euclidean distances using an UPGMA (unweighted pair group method with arithmetic mean) approach[32] (Fig. 1 and Supplementary Fig 2B). This way, we resolved many clusters of class 1 PBMs with distinct PDZ-binding propensities (Supplementary Fig 2B). We also identified a few additional clusters, implying PBMs of class 2, class 3 or mixed classes. Most interestingly, PBMs from HTLV1 Tax1 and HPV E6 were mostly clustered in a single clade (Figure S2B). This "oncoviral" clade also includes dozens of human motifs, that hence share common PDZ-binding preferences with the oncoviral PBMs. Most PBMs in this oncoviral clade share the class 1 consensus E-[TS]-x-V-COO⁻ (where x denotes for any residues), often immediately preceded by basic residue(s). These motifs showed generally high binding affinities with PDZ domains such as MAGI1_2, DLG1_2, SCRIB_1, SNTB1, or TX1BP3. The corresponding PDZ-proteins are often involved in maintenance of epithelial basolateral polarity[33]. Thus, interactomic distances may be particularly relevant to decipher viral hijacking, and the host motifs within the identified oncoviral clade likely represent an interactomic hot spot for oncoviral interference.

## Molecular basis of PBM recognition in the light of the quantitative interactome

For any individual PDZ or PBM measured in our quantitative interactome, binding specificities can be visualized by plotting binding profiles with p$K_d$ values sorted in decreasing order (Fig. 1, upper right panel or Fig. 3A, D). The steeper the profile slope, the more specific the PBM or PDZ for particular targets within the explored interactome[19,22]. The PBM-binding preferences of PDZ domains can also be visualized by calculating affinity-weighted sequence logos (see Methods). As compared to conventional logos built from unranked pools of binding sequences[34], affinity-weighted logos capture determinants of recognition specificity. Binding profiles and specificity logos can be generated and analyzed by users on the ProfAff server.

To illustrate the invaluable information provided by our affinity data and such modes of representation, we solved the crystal structures of several chosen PDZ-PBM complexes related to profiles and affinity-weighted logos of interest (see refinement statistics in

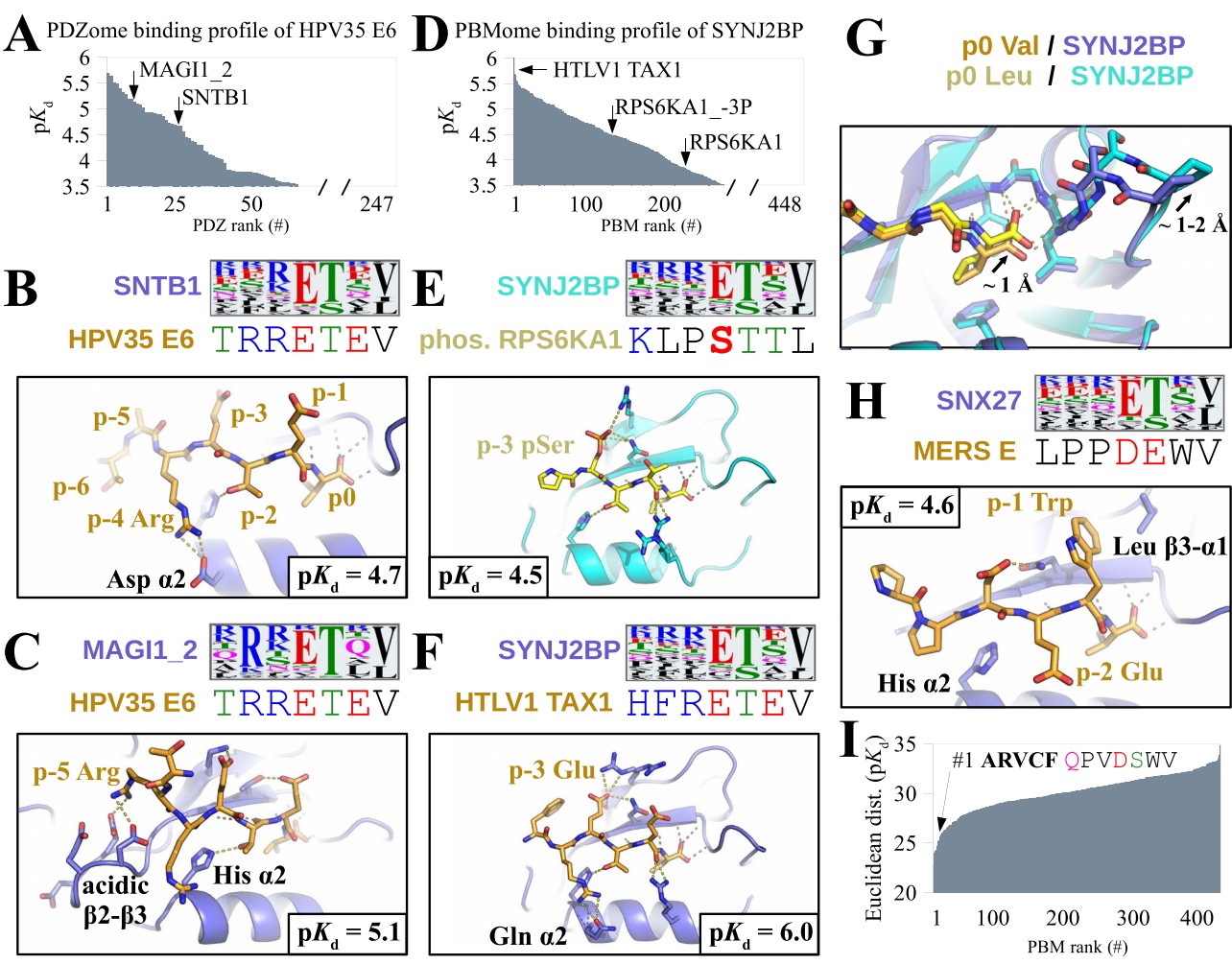

**Fig. 3 | Molecular determinants behind binding specificities in the PDZ-PBM interactome. A** PDZome-binding profile of HPV35 E6 PBM. **B** The affinity-weighted specificity logo of SNTB1 PDZ matches well with HPV35 E6 PBM at positions p0, p-2, p-3 and p-4, which display favorable contacts in the crystal structure of the SNTB1/HPV35 E6 complex. **C** MAGI1-2 PDZ logo matches with HPV35 E6 PBM at the same positions, but also at p-5 (Arg), exposed to the acidic β2-β3 loop of MAGI1_2 in the MAGI1/HPV35 E6 complex. Accordingly, HPV35 E6 binds stronger to MAGI1-2 than to SNTB1. **D** PBMome-binding profile of SYNJ2BP PDZ. **E** RPS6KA1 PBM matches poorly to the logo of SYNJ2BP. Accordingly, RPS6KA1 binds only weakly to SYNJ2BP. The p-3 phosphorylated RPS6KA1 PBM matches better to SYNJ2BP logo, with acidic phospho-Ser at p-3 contacting two residues of SYNJ2BP β2 and β3 strands in the complex. Accordingly, p-3 phosphorylated RPS6KA1 shows increased binding to SYNJ2BP. **F** HTLV Tax1 PBM sequence matches strongly with SYNJ2BP logo at p0, p-1, p-2, p-3, and p-4. Accordingly, Tax1 is the strongest SYNJ2BP binder in our interactome. In the SYNJ2BP-Tax1 complex, Glu at p-3 of Tax1 engages similar contacts as pSer of RPS6KA1_−3P, while Arg at p-4 provides additional contact to the α2 helix of SYNJ2BP. **G** Superposition of Tax1 and p-3 phosphorylated RPS6KA1 PBMs bound to SYNJ2BP. The carboxylate-binding loop of SYNJ2BP is shifted in the phosphorylated RPS6KA1-bound complex, most likely related to unfavorable Val → Leu substitution at p0 of RPS6KA1 PBM. **H** The logo of SNX27 indicates preference for class 1 PBMs (Ser/Thr residue at p-2). MERS E has a class 3 PBM (Glu at p-2) with poor match to SNX27 logo; yet MERS E binds relatively well to SNX27. In the SNX27/MERS E complex Trp at p-1 of MERS E establishes favorable hydrophobic contacts. **I** Interactomic distance profile of MERS E in the explored PBM space. The closest neighbor of MERS E is ARVCF, a class 1 PBM with Trp at p-1. Source data are provided as a Source Data file. For further details see Supplementary Figs. 2 and 3, Supplementary Table 1 and Supplementary Data 1. All logos were taken from the ProfAff server.

Supplementary Table 1 and crystallographic omit maps in Supplementary Fig 3). The logos of PDZ domains SNTB1 and MAGI1_2 indicate preferences for x-R-E-T-x-V-COO⁻ (Fig. 3B) and R-x-E-T-x-V-COO⁻ (Fig. 3C), respectively. The oncoviral PBM of HPV35 E6 satisfies both consensus requirements hence recognizes both PDZ domains (Fig. 3A). The structures of HPV35 E6 PBM bound to both SNTB1 and MAGI1_2 resolve their logo differences by revealing the distinctive contributions of Arg residues at p-4 and p-5 of HPV35 E6 in the two complexes (Fig. 3B, C).

In another example, the profile and the logo of SYNJ2BP indicate that this PDZ-domain is highly promiscuous, with however a mild preference for E-T-x-V-COO⁻ motifs and a moderate bias for basic residues at upstream (−4 and −5) positions (Fig. 3D–F). RPS6KA1 PBM does not fully match SYNJ2BP's logo (Fig. 3E) and thus binds only weakly to SYNJ2BP (pK_d = 3.82). Upon phosphorylation at p-3, RPS6KA1

gains a negative charge at this position, thereby getting closer to SYNJ2BP's logo. Accordingly, the p-3 phosphorylated variant RPS6KA1 gains affinity to SYNJ2BP (pK_d = 4.48; ΔpK_d = 0.66; Fig. 3D), as previously published[20]. The structure of p-3 phosphorylated RPS6KA1 bound to SYNJ2BP explains this enhanced binding by revealing a network of specific phosphoryl-PDZ contacts (Fig. 3E). Nonetheless, p-3 phosphorylated RPS6KA1 binds only mildly to SYNJ2BP as compared to SYNJ2BP's strongest binder in our interactome, the oncoviral PBM of HTLV1 Tax1 (pK_d = 6.04; Fig. 3D). In the structure of HTLV1 Tax1 PBM bound to SYNJ2BP, p-3 Glu of Tax1 engages contacts reminiscent of those of p-3 phospho-Ser of RPS6KA1, confirming SYNJ2BP's preference for a negative p-3 (Fig. 3F). Yet Tax1 matches better than phosphorylated RPS6KA1 to SYNJ2BP's logo thanks to a basic p-4 Arg and a p0 Val (Pro and Leu in RPS6KA1, respectively). Accordingly, in the crystal structures, p-4 Arg of Tax1 establishes favorable H-bonds with a

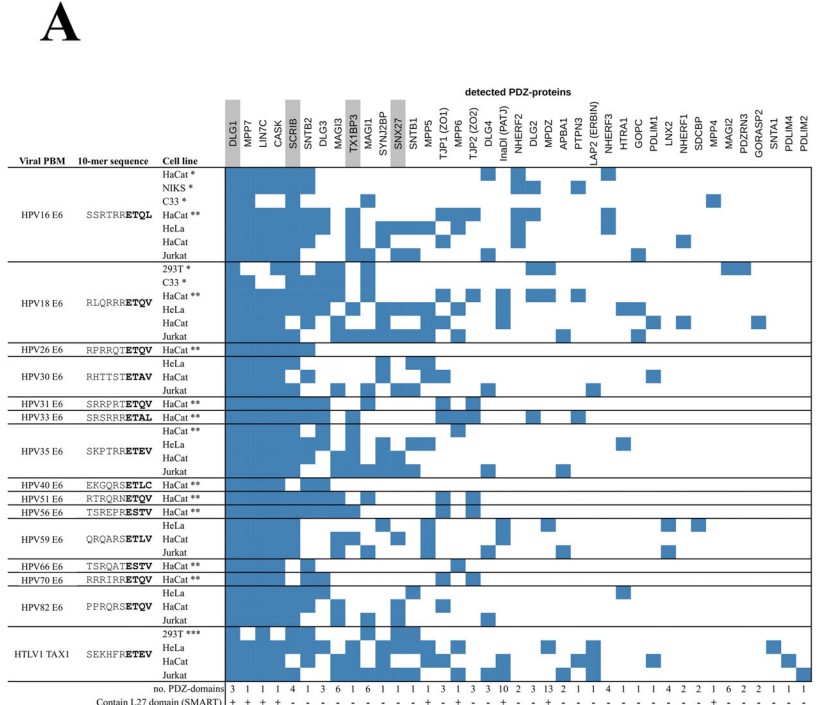

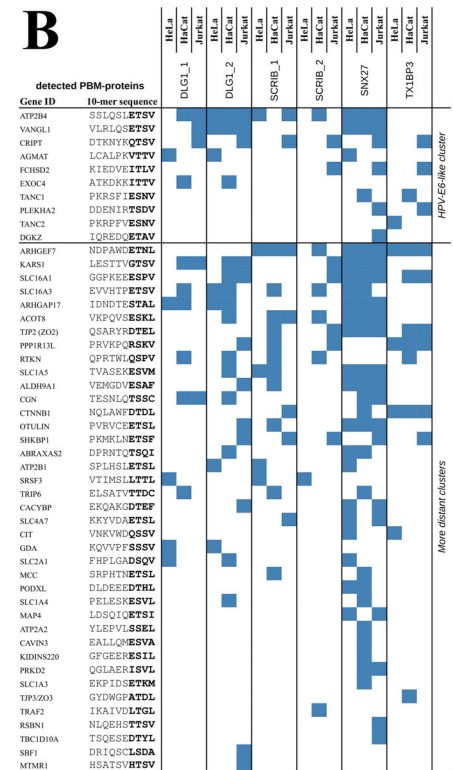

**Fig. 4 | Exploring interaction networks prone to perturbation by oncoviral PBMs in host cell models by AP-MS. A** Identification, from extracts of indicated cell lines, of PDZ-containing prey proteins retained by HPV E6 and HTLV1 Tax1 PBM baits. Results of comparable studies by Strickland et al.[39] (*) Thomas et al.[38] (**) and Al-Saleem et al.[40] (***) are also provided. For each PDZ-protein identified, the number of PDZ domains and the presence or absence of PDZ-oligomerization L27 domains are indicated at the bottom of the table. The four prey proteins containing the 6 PDZ domains that we selected as baits for the next round of AP-MS experiments are highlighted in gray. **B** Identification, from indicated cell extracts, of PBM-containing prey proteins retained by the 6 PDZ domains selected in **A**. The ten top

prey proteins of the list belong to the oncoviral-like human PBM clade revealed by our quantitative fragmental interactome. Interaction partners in **A** and **B** were found by comparing the prey quantities compared to a control resin. Binding threshold was defined at >2-fold enrichment and <0.01 *P* value, calculated by two-tailed unpaired T-test. Blue cells indicate detectable interaction between the immobilized bait (minimal binding fragment) and the endogenous prey (full-length protein). Note that only interaction partners containing PDZ domains or C-terminal PBMs are indicated. For a comprehensive list of all identified interaction partners, see Supplementary Data 2.

Gln from the α2 helix of SYNJ2BP PDZ, and the Val-bound state of SYNJ2BP PDZ is more compact than its Leu-bound state, due to displacement of the carboxylate-binding GLGF loop likely driven by the larger side-chain of Leu (Fig. 3G).

Specificity profiles and affinity-weighted logos also help analyzing interactions that fall out of general rules. According to its logo (Fig. 3H), SNX27 normally prefers class-1 PBMs, with Thr or Ser at p-2. Yet, SNX27 also binds ($pK_d = 4.56$) to the class-3 PBM of MERS E viral protein, with Glu at p-2. The structure of SNX27 bound to the PBM of MERS E shows a p-1 Trp of MERS E interacting with a hydrophobic groove of SNX27 including its β2-β3 sheets and a Leu side-chain from its β3-α1 loop. We found out that the closest interactomic neighbor of MERS E based on Euclidian distances of our explored interactome is a class-1 PBM, ARVCF (Fig. 3I). While ARVCF is a class 1 PBM with a p-2 Ser, it also possesses a Trp residue at p-1, similarly to MERS E. The p-1 Trp thus stands out as the major PDZ-binding specificity determinant for MERS E and ARVCF PBMs, more relevant than the class-determining p-2. Indeed, several PDZ-domains showed no clear class preference. For instance, HTRA1 can strongly bind to motifs from all three PBM classes. More generally, our interactomic distance analysis revealed several clusters of PBMs showing similar PDZ-binding patterns while belonging to distinct classes (Supplementary Fig 2B). While the nature of the antepenultimate p-2 residue is useful for classifying PBMs, it is not fully operative for predicting their PDZ-binding specificities.

## PDZ-PBM interaction networks perturbed by oncoviral PBMs in host cell models

While HPV and HTLV1 are both oncogenic viruses, they operate in distinct tissues, respectively epithelial cells[35] and T-lymphocytes[36,37]. Upon infection, their oncoproteins thus encounter distinct host proteomes. To investigate the influence of cellular context on oncoviral PBM-PDZ interactomes, we performed affinity purification mass spectrometry (AP-MS) from extracts of lymphoid Jurkat, keratinocyte HaCat and HeLa cells, respectively chosen as models of HTLV1 hosts, HPV hosts, and HPV18-transformed tumors.

Total cell extracts were first probed using seven PBM baits derived from HPV E6 of six distinct HPV types and HTLV1 Tax1 oncoproteins. 34 PDZ-protein preys were identified, whose identities are in excellent agreement with previous reports[38–40], (Fig. 4A, Supplementary Data 2). In particular, the oncoviral PBMs almost always captured DLG1, SCRIB, TX1BP3, and SNX27 from all cell types, all containing PDZ-domains displaying high affinities for these PBMs. PDZ-containing proteins CASK and MPP7 were also frequently pulled down, although their isolated PDZ domains do not detectably bind to E6 nor Tax1 in our PBM-PDZ interactome. Nonetheless, CASK and MPP7 contain hetero-tetramerization L27 domains[41]; these L27 domains can associate with L27 domains of PDZ-containing proteins that directly and detectably bind to E6 and Tax1, such as DLG or LIN7 members.

Next, the same cell extracts were probed using 6 PDZ-domains baits taken from 4 PDZ-proteins that interact ubiquitously with the oncoviral PBMs in all assayed cell lines, namely SCRIB, DLG1, TX1BP3,

and SNX27. We detected 73 enriched PBM-proteins, of which 10 belong to the closest neighborhood of E6 and Tax1 PBMs in our explored fragmentomic space (Fig. 4B, Supplementary Data 2). The 63 other enriched PBM-proteins mainly display class-1 motifs with sequences remarkably similar to those of E6 and Tax1 PBMs. Yet and similarly to the PBM-AP-MS experiments, the six PDZ domains also pulled-down, most likely indirectly, prey proteins devoid of C-terminal PBMs or exhibiting PBMs that did not detectably bind those PDZ domains in our PDZ-PBM interactome (see Supplementary Data 2 for a comprehensive list of all the interaction partners). For example, the E-cadherin and GIT1 proteins, both repeatedly identified by our PDZ-AP-MS assays, do not have any identifiable C-terminal PBM; they probably indirectly co-precipitate via their respective partners β-catenin (CTNNB1) and β-Pix (ARHGEF7)[42,43], both containing PBMs that detectably bound to PDZs in our quantified interactome.

In total, we identified 133 unique interactions between the 12 oncoviral PBM baits and 34 endogenous PDZ-protein preys, and 177 between the 6 PDZ baits and 73 endogenous PBM-protein preys. 57% of the PBM bait-PDZ prey interactions showed up in at least two cell types and 31% in all three types, whereas only 25% of the PDZ bait-PBM prey interactions showed up in at least two cell types, and less than 10% in all three types. Noteworthy enough, identification of partners primarily depends upon their expression levels, prone to vary across cell types. Considering all the experiments we performed, 53% of the identified PDZ-proteins but only 39% of the identified PBM-proteins were detected in at least two distinct cell types, and 32% of PDZ-proteins versus 19% of PBM-proteins were detected in all three cell types. This suggests that the host PDZ-proteins targeted by the oncoviral PBMs are more ubiquitously expressed than their host PBM-containing target proteins, and form a potential target group for oncogenesis independently of the host cell type.

Overall, these experiments largely cross-validate the fragmentomic data while illustrating that interactomics using AP-MS from cell extracts may generate false negatives (potential preys that are too weakly expressed) as well as false positives (indirect interactions). Remarkably, comparison to the fragmentomic interactome allows efficient curation of both issues.

## Capture of prey proteins by individual PDZ or PBM baits predominantly obeys the law of mass action

The enrichment of prey proteins on a given bait compared to a non-specific control is a key criterium for identifying interaction partners in AP-MS experiments[44]. The more enriched a protein, the more likely it is to be a bona fide interaction partner, suggesting proportionality between fold-enrichment values and amounts of prey-bait complexes formed, despite the washing steps commonly used in AP-MS. In our experiments, the fold-enrichment values of PDZ- or PBM-containing preys captured by particular PBM or PDZ baits in particular lysates were often strongly correlated to $pK_d$ values of the corresponding PDZ-PBM complexes, obtained from our quantitative interactome (Fig. 5A). We also observed strong correlations between fold-enrichment values of PDZ- or PBM-containing preys retained by the same PBM or PDZ baits across different cell extracts (Fig. 5B). The most straightforward interpretation of such enrichment-enrichment correlations across extracts is that enrichment values, in each extract, were correlated to common $pK_d$ values as in the instance shown in Fig. 5A. These observations suggest that the law of mass action, through which affinities dictate the proportions of complexes relatively to their free components, has a predominant impact on the outcome of AP-MS experiments using fixed amounts of baits, such as those we performed here. We however noticed that both types of correlations (enrichment-affinity and enrichment-enrichment) were overall weaker for experiments using PBM baits than those using PDZ baits (compare upper and lower right bar plots of Fig. 5A, B). While protein preys of PDZ baits generally contain only one C-terminal PBM, preys of PBM baits may

contain several PDZ domains, each displaying measurable affinity for the PBM. In such cases, we estimated affinities by assuming additivity of association constants, as proposed by others before[45] (see Methods). This approximation may contribute to the weaker correlations observed for some experiments using PBM baits.

## Proteomic perturbation upon expression of a viral PBM-containing oncoprotein

We chose HPV16 E6 as a model to investigate how the PBM of a viral oncoprotein can alter the molecular physiology of host cells at a system-wide level, beyond its direct interaction partners. E6 is built upon a core folded region composed of two zinc-binding repeats followed by a short intrinsically unfolded C-terminus bearing its PBM[46–48]. As shown in the preceding results, the E6 PBM targets several ubiquitous PDZ proteins and potentially perturbs a large set of host PBM proteins belonging to the "oncoviral" clade. However, HPV16 E6 has many other partners beyond PDZ-proteins[49,50]. In particular, E6 forms a trimeric complex with the "LxxLL" motif of ubiquitin-ligase E6AP[47] and the core domain of tumor suppressor p53[51], leading to ubiquitinylation and proteasome-mediated degradation of both proteins[51,52] in a PBM-independent manner[53].

We measured the proteomes of HEK293T cell-lines stably expressing wild-type HPV16 E6 or HPV16 E6ΔPBM, a mutant devoid of the C-terminal PBM. As compared to a control vector, both E6 and E6ΔPBM induced a proteome-wide perturbation affecting ~10% of 2273 human proteins detected in all three conditions, demonstrating a specific PBM-independent impact of E6 core region (Fig. 6A). Among these significantly perturbed proteins, a common group of 104 proteins (~5% of the detected proteome) showed changes of abundance in the same general direction, with a Pearson correlation coefficient (PCC) of 0.85. This group seems to be involved in biological processes such as viral carcinogenesis or translational control as calculated by GO-enrichment analysis (Supplementary Data 3). In particular, the abundances of both E6AP and p53 strongly decreased in the presence of E6 or E6ΔPBM, with comparable efficiency (Fig. 6A, D, E).

Nonetheless, we also found extensive PBM-dependent effects. Between E6- and E6ΔPBM-expressing cells, 191 proteins (8% of the observed proteome) displayed a significant change in abundance and this change exceeded a two-fold ratio in the case of 69 proteins (3% of observed proteome) (Fig. 6B). 50 (72%) of those 69 markedly altered proteins varied in opposite directions in HPV16 E6 or E6ΔPBM cells compared to the control cell line (Fig. 6C). For example, the concentration of the E3 ubiquitin-protein ligase HUWE1 is decreased by threefold by E6 yet increased by twofold by E6ΔPBM (Supplementary Data 3). In the remaining 28% of cases, both E6 and E6ΔPBM caused a concentration variation in the same direction, with the PBM significantly boosting or softening the effectiveness of the E6 core region.

HPV E6 may target some PDZ-proteins to degradation[13]. In our data, PDZ-proteins do not particularly stand out among proteins differentially affected by the E6 PBM. Among 11 PDZ-proteins detected in our proteomic survey, no significant change was observed between E6- and E6ΔPBM-transformed cells (Fig. 6D). Since the remaining ~140 known human PDZ-proteins were undetected by the MS approach, we used western blot to further investigate the amounts of five of the main E6-binding PDZ proteins (SCRIB, MAGI1, DLG1, SNX27, NHERF3) (Fig. 6E and Supplementary Fig 4A). As compared to control cells, the only significant change was observed for SCRIB levels, which decreased by ~30% in full-length E6-expressing cells, while upon expression of E6ΔPBM SCRIB concentration was unchanged. We also quantified the mRNA levels of several PDZ-proteins in our stable cell lines with RT-qPCR, without observing any significant changes (Supplementary Fig. 4B). Therefore, the PBM-dependent decrease of SCRIB levels in cells expressing full-length E6 likely reflects a shorter protein lifetime. These variations of SCRIB levels still remain quite moderate when compared to the striking decrease in p53 and E6AP levels

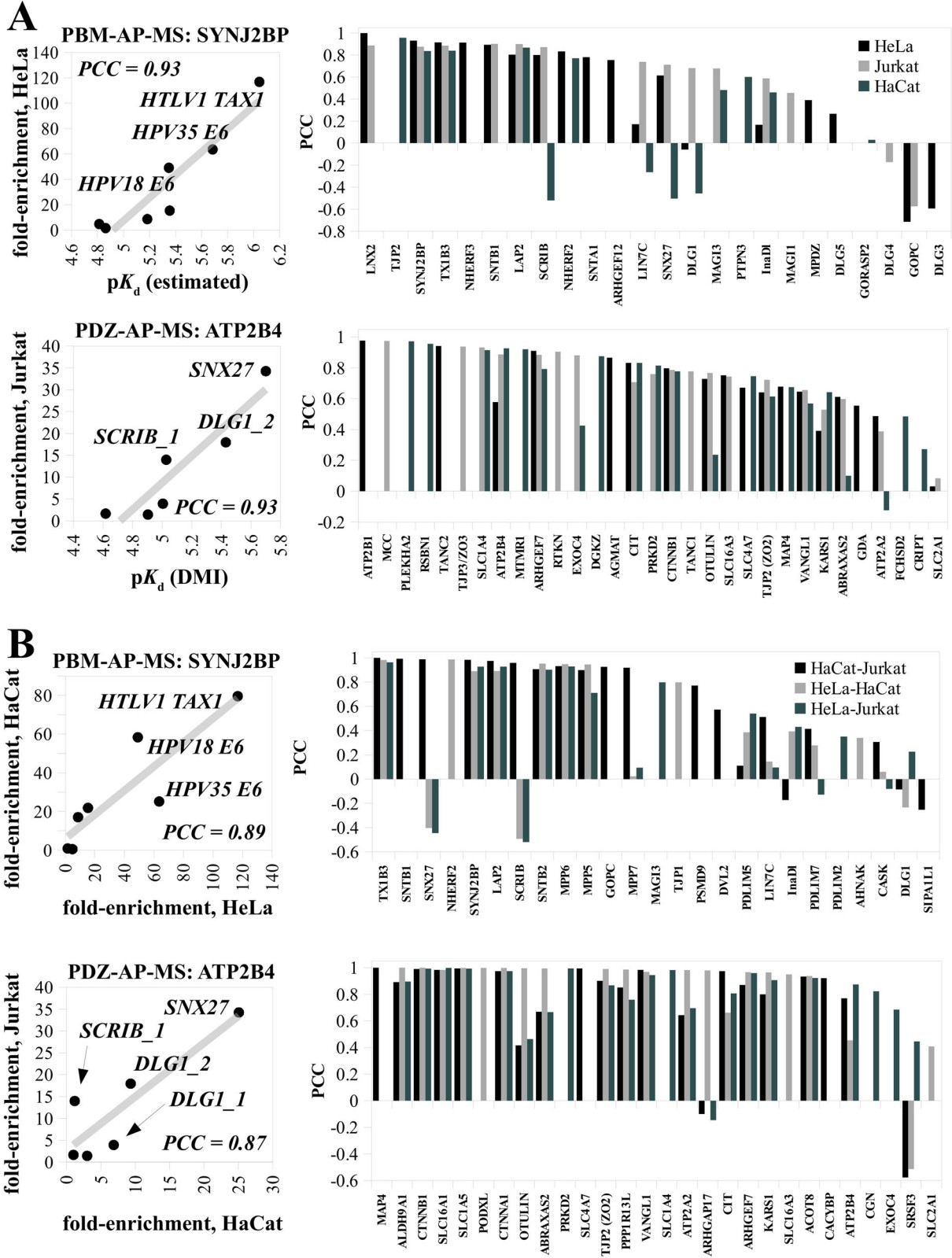

induced by the core region of E6 independently of its C-terminal PBM (Fig. 6D, E).

## Discussion

Here we systematically performed 65,000 distinct individual affinity measurements in a defined region of the human interactome. We focused on minimal protein interaction fragments. For affinity-based interactomics, these essential building blocks of the proteome present both theoretical and practical advantages over full-length proteins. The 24,000 human proteins constitute *at minimum* a binary inter-actomic space of about 500,000,000 pairs (24,000²), which already exceeds by 100 to 1000-fold the numbers of human protein-protein interactions documented in Biogrid (~500,000), HuRI (~56,000), Bio-Plex (~120,000) or PCP-SILAM (~125,000)[2,3,30,54]. Furthermore, full-

**Fig. 5 | Amounts of complex formation between endogenous preys and bait fragments in AP-MS correlate with their corresponding affinities.**
**A** Correlations between enrichment values and affinities measured using PBM baits (upper panels) or PDZ baits (lower panels). For preys containing several PDZ domains, we assumed additivity of association constants (see Methods). On the left side, example correlations are shown for one PDZ (SYNJ2BP) and one PBM (ATP2B4) prey protein. On the right side, only the Pearson correlation coefficient (PCC) is shown for measured preys. Note that correlations are overall weaker than

in the case of PDZ preys (upper panels), with an average PCC of 0.5 instead of 0.8. **B** Correlations between enrichment values obtained from different cell lysates measured using PBM baits (upper panels) or PDZ baits (lower panels). On the left side, example correlations are shown for one PDZ (SYNJ2BP) and one PBM (ATP2B4) prey protein. On the right side, only the Pearson correlation coefficient (PCC) is shown for measured preys. Note that correlations are overall weaker than in the case of PDZ preys (upper panels). Source data are provided as a Source Data file. See Supplementary Data 2 for more details.

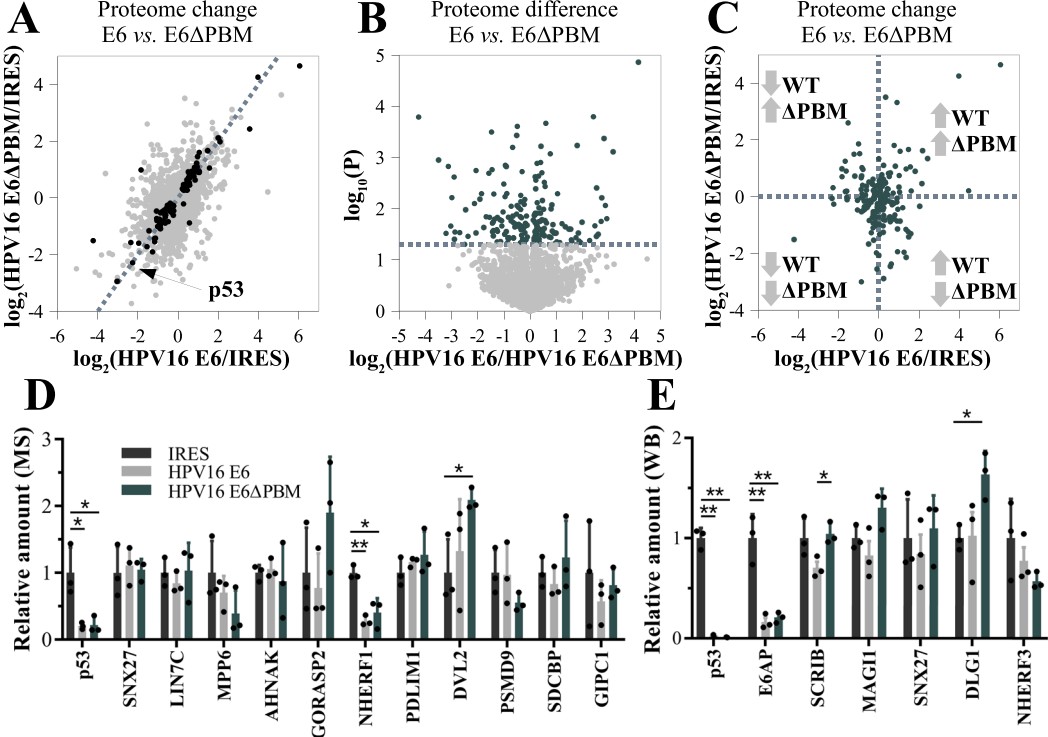

**Fig. 6 | Proteomic impact of stable HPV16 E6 expression in HEK293T cells and the contribution of its PBM. A** Proteomic changes measured with label free mass spectrometry ($n = 3$) in HEK293T cells stably expressing HPV16 E6 or HPV16 E6ΔPBM compared to a control cell line "IRES". Dots indicate the detected 2273 proteins. Black dots indicate proteins whose abundances changed significantly in both E6- and E6ΔPBM-expressing cells as compared to control cells. The dotted line indicates the diagonal. **B** Proteomic difference between the cell lines expressing E6, or E6ΔPBM. The dotted line indicates the statistical significance threshold ($P < 0.05$, two-tailed unpaired T-test) and dark dots indicate proteins whose statistical difference is higher than this threshold. **C** Proteins that show statistically significant differences in concentration between E6- and E6ΔPBM-expressing cell lines often

change in opposite direction compared to the control cell line. **D** Relative amounts of 11 PDZ proteins and p53 in the three cell lines measured by the proteomic survey. Relative XIC intensities are shown compared to the intensities of the control cell line. **E** Relative amounts of 5 PDZ-proteins, as well as p53 and E6AP, quantified by western blot. GAPDH normalized abundances were compared to the intensities of the control cell line. In **D** and **E**, mean and STD were calculated from three individual measurements indicated by black dots, and statistical significance is indicated by * for $P < 0.05$ and ** for $P < 0.01$. Two-tailed unpaired t-tests were used to determine statistical significance. Exact $p$ values are provided in the source data file together with the other source data. See Supplementary Fig 4 for results of western blots and Supplementary Data 3 for full proteomic results.

length proteins can exist as many diverse conformational and/or chemical proteoforms[55], arising from RNA splicing, post-translational modifications, or binding to third-party proteins or other ligands. All those proteoforms may have distinct binding properties[56] and they all participate in biological function. Therefore the full human protein interactome includes many billions of proteoform-proteoform affinities. If one adds the practical difficulty of distinguishing all the different proteoform pairs, the exhaustive measurement of all these affinities seems definitely unachievable. In contrast to full-length proteins, a well-defined minimal interacting fragment pair usually displays one single affinity constant. The human proteome is estimated to contain 35,000 domain instances dispersed across 10,000 distinct families and about one million motifs, including post-translational variants[8]. The full fragmental interactome, once subdivided in its multiple domain-motif families, should be limited to a few hundreds of millions of interactions. We covered here more than 6% of the

1,000,000 potential affinities of one of the largest domain-motif interactomes in human. This was achieved at a pace of up to 10,000 affinity measurements per day, using mainly a benchtop protocol and very limited material resources. Taken up by a collective research initiative using robotized instrumentation, this pace can be increased manyfold, putting the exhaustive affinity mapping of the full human fragmental interactome within realistic reach. Once measured, fragmental affinities could then be combined with proteome-wide structure modeling tools such as *Alphafold*[57], for the theoretically challenging yet exciting structure-guided prediction of binding properties of all interactomic combinations of multiple proteoforms.

We quantified 18,000 PDZ-PBM affinity constants, representing 28% of the explored interactome. Yet the remaining 47,000 pairs that fell below holdup quantification threshold should not be considered as "non-interactions". Any pair of molecules, even the least compatible with each other, displays an affinity constant reflecting their

probability of encounter. The proportion of quantified constants only depends on the sensitivity of the measurement. Furthermore, within the PDZ-PBM space, even unfavorable pairs retain elementary features for mutual recognition. We have previously crystallized unfavorable PDZ-PBM complexes whith affinities below holdup threshold[21]. The PBMs were nonetheless visualized and located on the usual PDZ pocket in the crystal structures. Their affinities, measured by alternative methods, were in the millimolar range, not far from our current holdup $pK_d$ threshold comprised between 4 and 3.1. Therefore, a more sensitive approach reaching out for the millimolar range might have sufficed to quantify a much larger part of the 65,151 affinities measured in the present study. Measuring all affinities down to the weakest ones is not a superfluous task. While the few strongest interactions may look more "functionally relevant" by their capacity to disrupt the system when they are missing or altered, the numerous weak interactions also contribute to homeostatic function[58,59]. This may be particularly relevant for promiscuous interactomes such as the PDZ-PBM space, where the strongest interactions remain confined to a relatively narrow affinity range ($5 < pK_d < 6$) while being greatly outnumbered by weak interactions, which may thus collectively contribute to a significant fraction of the complexes formed.

Our AP-MS experiments showed that amounts of detected complexes vary depending on the proteomic content of cells and are governed by affinities. Based on this observation already hinted in our previous work[60], we propose to distinguish "interactomes", viewed as intrinsic interaction propensities of biological systems, from "complexomes", viewed as extrinsic interactions occurring in particular contexts. Interactomes are quantifiable by binding constants and complexomes by concentrations of complexes. By generalization of the law of mass action, the quantitative interactome of an organism contains the information that relates expressed proteomes to complexomes, in different cellular or sub-cellular states of that organism. Under this view, the two hybrid resource HuRI[2] is a qualitative full-length protein interactome; the AP-MS resource BioPlex[3] and the co-fractionation resource PCP-SILAM[54] are both qualitative protein complexomes; and ProfAff (present work) is a quantitative fragmentomic interactome.

Quantitative fragmentomics provide access to a yet unexplored multidimensional interactomic space, where affinity-based distances constitute the ultimate way, superior to sequence-based prediction, to situate and compare molecular actors of a biological system. This turns out to be particularly useful for analyzing viral hijacking. We identified an interactomic host spot for oncoviral interference, targeted by both HPV E6 and HTLV Tax1 viral oncoproteins despite their distinctive epithelial and lymphoid tropisms. This hot spot comprises a core of ubiquitously expressed PDZ-proteins, and PBM-proteins that vary more across cell types, as supported by AP-MS experiments. To investigate how interactome-interfering properties of a viral PBM can perturb cellular physiology, we studied the impact of the PBM of HPV oncoprotein E6 on the proteomic profile of E6-expressing cells. While, as expected, E6 induced a dramatic PBM-independent decrease of p53 and E6AP levels, it induced a moderate PBM-dependent decrease of only one detected PDZ-protein, SCRIB. Yet, PBM-dependent E6-induced perturbations impacted numerous non-PDZ proteins, representing 8% of the overall detected proteome. This points to distinct mechanisms of E6 action over p53 and E6AP, or on PDZ proteins. For p53 and E6AP, E6 employs a reductionist approach by plugging them to the ubiquitination system[52]. Since both p53 and E6AP are transcription factors[61,62] and E6AP is a E3 ubiquitin ligase with genome-wide impact[63], their degradation expectedly induces system-wide transcriptomic and proteomic changes. With PDZ proteins, E6 rather employs a holistic approach by perturbing an intricated network of promiscuous transient interactions, which also ends up provoking system-wide proteomic changes. In our quest to understand cellular life, we may gain to inspire from the balanced reductionist and holistic approach of papillomaviruses.

## Methods

### Synthesis of purified biotinylated and fluorescent peptides used for holdup, fluorescence polarization, and AP-MS experiments

Peptides for holdup assay systematically incuded a N-terminal biotin, covalently added to their N-terminus via a chemical linker. We have previously extensively used biotinylated peptides for binding assays[19,22,64]. In our experience, we never found that the inclusion of a N-terminal linker-biotin moiety significantly altered the structure nor the binding properties of protein-peptide complexes.

HPLC-purified (all >95% purity) biotinylated peptides were chemically synthesized on an ABI 443A synthesizer with standard Fmoc strategy with biotin group attached to the N-terminus via a TTDS (Trioxatridecan-succinamic acid) linker, or were commercially purchased from Genicbio (Shanghai, China) with biotin group attached to the N-terminus via an Ahx (6-aminohexanoic acid) linker. Predicted peptide masses were confirmed by mass-spectrometry. Peptide concentrations were determined based on their dry weight.

Fluorescent peptides f16E6: fluorescein-RTRRETQL; fRSK1: fluorescein-KLPSTTL and fpRSK1: fluorescein-KLPpSTTL were chemically synthesized on an ABI 443A synthesizer and HPLC purified with fluorescein coupled directly to the N-terminus. The biotinylated MERS-E peptide was FITC labeled (fMERS: biotin-Ahx-DSK(-fluorescein) PPLPPDEWV) as follows: the peptide was mixed with sub-stoichiometric FITC (Sigma-Aldrich, St. Louis, MO, USA) in a basic HEPES buffer (pH 8.2). Labeling reaction was stopped with 100 mM TRIS and the reaction mixture was buffer exchanged in order to remove fluorescent contaminants. The concentrations of fluorescent peptides and labeled fraction of fMERS were determined based on their dry weight and their fluorescence intensity. For competitive fluorescence polarization assay, only HPLC purified peptides were used.

### Reagents for crude peptide library synthesis

All solvents and chemicals were used without any further purification and purchased as peptide grade when available. N-methylpyrrolidinone was purchased from Biosolve. Diisopropylcarbodiimide and collidine were purchased from Sigma. The following L-amino acids were used: Fmoc-Ala, Fmoc-Arg(pbf)-OH, Fmoc-Asn(Trt)-OH, Fmoc-Asp(Otbu)-OH, Fmoc-Gln(Trt)-OH, Fmoc-Glu(Otbu)-OH, Fmoc-Glu-Otbu, Fmoc-Gly-OH, Fmoc-His(Trt)-OH, Fmoc-Ile-OH, Boc-Ile-OH, Fmoc-Leu-OH, Fmoc-Val-OH, Fmoc-Lys(boc)-OH, Fmoc-Ser(Otbu)-OH, Fmoc-Thr(Otbu)-OH, Fmoc-Tyr(Otbu)-OH, FmocTrp(Boc)-OH, Fmoc-Phe-OH, and they were purchased from either Protein Technologies, Novabiochem Merck or IRIS Biotech (Germany). Biotin was purchased from Sigma. Fmoc-preloaded resin was used and purchased from Merck Millipore. Fmoc-8-amino-3,6-dioxaoctanoic acid (Fmoc-Ado-OH) was purchased from Flamma Group, Italy.

### Crude peptide library synthesis used for holdup experiments

Crude peptide libraries were synthesized by parallel 96 format peptide synthesis using Intavis multipep RSi. Filter plates from NUNC (Thermo-Fischer) were used loaded with 10 mg per well of Fmoc-preloaded polystyrene resins (Merck Millipore). Fmoc-amino acids (Fmoc-AA) as 0.3 M in dimethylformamide (DMF) containing 0.3 M OxymaPure were used. Three consecutive couplings using 100 µl Fmoc-AA and 11 µl DIC (3 M in DMF) were employed with coupling times of 5, 15, and 60 min. Removal of Fmoc was done by washing twice with 120 µl of 25% piperidin in DMF for 2 and 8 min. Washing was done by adding 150 µl to each well by the 8-pin manifold five times. Coupling of biotin was done as a 0.3 M solution in DMSO containing 0.3 M OxymaPure and activated by DIC. Coupling time was extended to 15 min, 45 min,

and 360 min. TFA cleavage was done by 93% TFA containing 4% triisopropylsilane 1% thioanisol 3% $H_2O$ for 2 h. The total volume of 1 ml TFA added was added to each well. TFA was reduced in volume to -150 µl followed by precipitation by the addition of diethyl ether. The peptides were transferred to Waters solvinert plates and washed thoroughly five times with diethyl ether. After washing with diethyl ether a small fraction of the peptide slurry was added to a microtiter plate and dried followed by UPLC-MS analysis after solubilization in DMF/$H_2O$ (1:1). Analysis of purity and identity by MS was performed using Waters Acquity UPLC system, with Waters Acquity TUV detector 214 nm connected to a LCT Premier XE mass spectrometer from Micromass. The buffer used were 0.05% TFA in $H_2O$ (buffer A) and 0.05% TFA in acetonitrile (buffer B) using a gradient of 5–60% B for 3.5 min on a Waters column BEH C18 1.7 mm; 2.1 mm × 50 mm and a flow rate of 0.45 ml/min.

All peptides had a biotin group attached to the N-terminus via an Ado-Ado linker. Internal standards used for fluorescent holdup normalization (see below) were re-synthetized as crude peptides. No apparent differences in these internal standards were observed between HPLC purified and crude peptides in holdup experiments. Predicted peptide masses were confirmed by mass-spectrometry in all cases and the purity was found to observably reach >90% in most cases. Average peptide concentrations were determined based on the excess dry weight of the entire 96-well plate and we used 10–50× molar excess (based on dry weight) in holdup experiments to take into account the variability of peptide-to-peptide yields.

### Preparation of PDZ library in bacterial lysates for holdup experiments (by LYSC)

The PDZome v2 library, consisting of all 266 human PDZ domains as $His_6$-MBP-PDZ constructs, was prepared as previously described[27]. We have previously used MBP to facilitate and standardize the production of many recombinant proteins and domains in a folded, soluble and active form[19,47,60,64,65]. We never found that the presence or absence of MBP significantly altered the structure nor the binding properties of protein-peptide pairs. However, it remains statistically possible that the affinities of some particular PDZ-PBM pairs addressed in our screen have been altered by the inclusion of MBP.

The library was expressed in *E. coli* BL21(DE3) with an auto-induction media and was lysed as described elsewhere in detail[27]. The $His_6$-MBP-PDZ concentrations of soluble cell lysate fractions were evaluated with a microfluidic capillary electrophoretic system (Caliper LabChip GXII, PerkinElmer, Waltham, Massachusetts) and were adjusted to 4 µM by dilutions. Lysozyme and BSA internal standards were added to the library before freezing the library in 96 well plate for the holdup experiment.

### Single affinity purified protein expression and purification for holdup experiments (by SAPF)

For single affinity purification, the PDZome v2 library was expressed in the same conditions (strain, media, buffers) than for the bacterial lysate[27] but at a bigger culture scale (24 ml per PDZ instead of 6 ml per PDZ). The production (24 ml) and Ni-affinity purification (800 µl of beads/PDZ) follow strictly the protocol described in ref. 66. The elution volume of Ni-affinity purification was reduced from 4 ml[66] to 2.5 ml/PDZ. 96 PDZ are produced in parallel and are purified in four blocks. At the end of the purifications of these blocks purified PDZs are desalted using Zeba Spin plates (Thermo scientific) in 50 mM Tris, pH 8.0, 300 mM NaCl, 10 mM imidazole. The purity and concentration of each $His_6$-MBP-PDZ were evaluated with a microfluidic capillary electrophoretic system (Caliper LabChip GXII, PerkinElmer, Waltham, Massachusetts) and adjusted to 4 µM by dilutions before freezing the library in 96-well plate for the holdup experiment.

### Double affinity purified protein expression and purification for holdup experiments (by DAPF) and fluorescence polarization assays

For double affinity purification, $His_6$-MBP-PDZ constructs from the PDZome v2 library[27] were expressed in *E. coli* BL21(DE3) with IPTG induction (1 mM IPTG at 25 °C for 4−5 h) and harvested cells were lysed in a buffer containing 50 mM TRIS pH 7.5, 150−300 mM NaCl, 2 mM BME, complete EDTA-free protease inhibitor cocktail (Roche, Basel, Switzerland), 1% Triton X-100, and trace amount of DNAse, RNAse, and Lysozyme. Lysates were frozen at −20 °C before further purification steps. Lysates were sonicated and centrifuged for clarification. Expressed PDZ-domains were captured on pre-packed Ni-IDA (Protino Ni-IDA Resin, Macherey-Nagel, Duren, Germany) columns, were washed with at least ten column volume cold wash buffer (50 mM TRIS pH 7.5, 150 mM NaCl, 2 mM BME) before elution with 250 mM imidazole. The Ni-elution was collected directly on a pre-equilibrated amylose column (amylose high flow resin, New England Biolabs, Ipswich, Massachusetts). Amylose column was washed with five column volume cold wash buffer before fractionated elution in a buffer containing 25 mM Hepes pH 7.5, 150 mM NaCl, 1 mM TCEP, 10% glycerol, 5 mM maltose, complete EDTA-free protease inhibitor cocktail. The concentration of proteins was determined by their UV absorption at 280 nm before aliquots were snap freeze in LN2 and storage at −80 °C.

### Holdup assay

**Principle of the holdup assay and previously published versions.** The holdup assay is a comparative chromatographic retention approach devoid of washing steps, which in contrast to pull-down assay allows monitoring the steady-state binding equilibrium between one molecular species attached to resin and another one present in solution[28]. Streptavidin (GE Healthcare, Uppsala, Sweden) resin is saturated with either biotin ("control resin"), or a biotinylated peptide. Then, a PDZ-containing solution is added to both resins and the reaction mixture is agitated for 15 min at room temperature. The whole experiment is carried out on filter plates (Millipore, Burlington, Massachusetts) and the incubation step is stopped with rapid filtration to separate the unbound PDZ fraction. Finally, the concentration (a.k.a. intensities) of $His_6$-MBP-PDZ is determined and binding intensities (BI) are calculated using Eq. (1):

$$BI = \frac{I_{total} - I_{depleted}}{I_{total}} \qquad (1)$$

where $I_{total}$ is the total intensity (e.g., concentration, peak intensity, fluorescent signal, etc.) of the PDZ present in the flow-through of the biotin-saturated control resin, and $I_{depleted}$ is the intensity of PDZ in the flow-through of the peptide-saturated resin. In the holdup buffer (50 mM TRIS pH 7.5, 300 mM NaCl, 1 mM TCEP) at least a single internal standard was used (BSA/lysozyme or fluorescein/mCherry) for peak intensity normalization.

Previously, the holdup approach was automatized on liquid handling systems[19]. In this method, a total bacterial lysate over-expressing the $His_6$-MBP-PDZ protein was used, and the $His_6$-MBP-PDZ intensities were measured using capillary electrophoresis. While the ease of use of such a complex matrix is desirable (ease of library preparation, crowded environment, etc.), the readout is both slow and requires a multi step data-curation, as described in details in refs. 22,67. To eliminate this bottleneck, we also developed an intrinsic fluorescence based holdup method, where the concentration measurement is done based on the fluorescence of Trp residues of purified $His_6$-MBP-PDZ constructs[64]. The fluorescent readout is fast and can be done on any fluorescent plate-reader, however at a cost of accuracy. Indeed, any contaminant present in the holdup mixture, such as contaminants from bacterial source, or spontaneously cleaved $His_6$-MBP, will generate a background fluorescence, that will decrease the

observable partial activity of each PDZ sample. The real partial activity of each sample can be retrieved through calibration with some BI values obtained previously by capillary electrophoresis-based holdup.

**New developments increase the throughput and the precision of the holdup method.** Automated holdup experiments were originally designed in a way where 96 different PDZ domains were tested against 4 wells on a 384 well filter plate, 3 different/identical peptides, and a single control. In a reverse holdup layout, multiple peptides are tested against the same protein. For example, 370 peptides and 14 biotin controls can be placed on a single 384 well filter plate and then tested against a single PDZ. This way, not only the number of tested peptides increase, but also the number of controls, too. In previous holdup experiments, the singlicate control for each PDZ caused an under-determined situation where the precision of the determined BI value was mainly determined by the accuracy of the single control measurement. For example, if an interaction was reportedly measured in triplicates, three peptide-saturated wells were compared with a single control. In the reverse holdup layout, many control wells are measured on the same PDZ and therefore the total intensity can be determined at high precision and great accuracy. For example, if an interaction was reportedly measured in singlicate, each peptide saturated well was compared with 10–15 controls.

As a minor tweak, we also prepared presaturated resin stocks for the reverse layout holdup experiments, instead of saturating resins directly on the filter plate. This allowed us to achieve a higher reproducibility by minimizing the well-to-well variability of resin immobilization rates and to prepare multiple plates simultaneously. Stocks were prepared by simply up-scaling the described procedure for a single well (see below). Stocks were stored in holdup buffer and the biotin depletion step (where the nearly peptide/biotin saturated resins are incubated with large excess of biotin to deplete the remaining vacancies in the resin) was only executed after transferring the desired amounts to filter plates.

Due to the extremely slow dissociation rate of the biotin-streptavidin interaction, it is possible to recycle the holdup-treated resins in the filter plate several times (at least up to 20-25 times from our personal experience, the holdup measurements being highly reproducible at least up to this number). Recycling was carried out in 5 steps. The plates were washed first with ten resin volume 1 M NaCl, then with ten resin volume 2 M Urea, and finally, three times with ten resin volume holdup buffer. All steps in the holdup protocol where the liquid fraction needs to be filtered can be performed either on a vacuum manifold, or by centrifugation with the exception of the last washing step before adding the PDZ-solution and the holdup reaction itself, which both need to be performed by centrifugation.

By combining the fluorescent readout, the reverse holdup layout, and the resin recycling, not only the throughput of the holdup assay increased by several orders of magnitude, but also its precision and accuracy. In addition, this optimized protocol does not require any specialized instrument and can be implemented with a simple plate reader, a centrifuge, and a multichannel pipette. Optimally, with the help of a vacuum manifold, an electric multichannel pipette (such as an E1-ClipTip pipette from Thermo Fisher, Waltham, Massachusetts), and a liquid dispensing station (such as a Flexdrop IV from PerkinElmer, Waltham, Massachusetts), the holdup assay is capable to measure the affinities of 5–10,000 interactions a day at a fraction of the cost of the originally developed automatic setup.

**Interaction measurements with holdup experiments.** In the present work, we include data from three types of holdup measurements: holdup measurement with bacterial lysates coupled with capillary electrophoretic readout (LYSC), holdup measurement with high-throughput single affinity purified PDZ protein coupled with

fluorescent readout (SAPF), and holdup measurement with double affinity purified PDZ protein coupled with fluorescent readout (DAPF).

Most LYSC data originate from previous articles[19–22,26,68,69]. Additional LYSC holdup experiments were carried out as previously described, using only the original layout. Briefly, 2.5 µl streptavidin resin was mixed with biotin or biotinylated peptide at 55 µM concentration in 6.5 resin volume for 15 min to achieve resin saturation, then after a single washing step (ten resin volume, holdup buffer), the resin was depleted with biotin (1 mM biotin, 5 resin volume, 15 min). Holdup experiments were carried out after three washing steps (ten resin volume, holdup buffer). 5 µl bacterial lysate with 4 µM His$_6$-MBP-PDZ supplemented with 4 µM BSA and 0.05 mg/ml lysozyme was incubated on the filter plate for 15 min with shaking before filtration by centrifugation. Filtrates were analyzed with Caliper Labchip GXII (Perkin Elmer, Waltham, Massachusetts), and collected data were analyzed with in-house developed algorithms[22,67]. In total, our database includes 4475 unique interactions characterized by CALIP method, originating from 9920 measurements. We have found that the standard deviation of duplicate experiments ($n = 1893$) was $\sigma = 0.05$ BI. We set our binding detection threshold based on this and defined it at $2\sigma$ (BI = 0.1). Based on this threshold, 807 unique interactions showed detectable binding in CALIP holdup experiments.

SAPF experiments were carried out in the original layout. Resin preparation was identical as in LYSC holdup experiments. In each well 2.5 µl peptide- or biotin-saturated resin was incubated with 5 µl single affinity-purified His$_6$-MBP-PDZ at a concentration of 4 µM. His$_6$-MBP-PDZ was supplemented with 50 nM fluorescein and 100 nM mCherry. Incubation was carried out on filter plates for 15 min with shaking before filtration. Filtrates were analyzed on a PHERAstar (BMG Labtech, Offenburg, Germany) microplate reader by using $485 \pm 10$ nm–$528 \pm 10$ nm (fluorescein), $575 \pm 10$ nm–$620 \pm 10$ nm (mCherry), and $295 \pm 10$ nm–$350 \pm 10$ nm (Trp-fluorescence) band-pass filters. In total, our database includes 6850 unique interactions characterized by SAPF method, all measured in singlicate. Due to the lack of replicates, we set the same binding detection threshold that we defined for LYSC experiments (BI ≥ 0.1). Based on this threshold, 1101 unique interactions showed detectable binding in SAPF holdup experiments.

DAPF experiments were carried in the reverse layout with presaturated resins. Resin presaturation was performed with the same volume/concentration ratio as in the resin preparation for the LYSC and SAPF experiments, but over a longer period of incubation (2–4 h). The presaturated resin slurry was supplemented with 20% ethanol for longer-term storage. After distributing 5 µl presaturated streptavidin resin on filter plates, after a single washing step (10 resin volume, holdup buffer), the resin was depleted with biotin (1 mM biotin, 5 resin volume, 10 min). After two washing steps, the resin was incubated with 10 µl double affinity-purified His$_6$-MBP-PDZ solutions, which was supplemented with 50 nM fluorescein and 100 nM mCherry. Fluorescence intensity measurements were performed as described in SAPF experiments. The reverse layout made it possible to include several peptides on the filter plate that was measured previously with Caliper-based holdup methods. In case the reverse layout was used, measured BI values were compared to LYSC standards to determine the partial activity of each PDZ sample. Then, the determined partial activity was applied in the calculations to normalize the obtained BI values. In total, our database includes 59,179 unique interactions characterized by DAPF method, originating from 62,604 measurements. We have found that the standard deviation of duplicate experiments ($n = 2780$) had an average of 0.029 BI (with a few outliers) and a median of 0.019 BI, respectively. Based on these, we used $\sigma = 0.025$ to define our binding detection threshold at $2\sigma$ (BI = 0.05). Based on this threshold, 17,726 unique interactions showed detectable binding in DAPF holdup experiments.

## Competitive fluorescence polarization (FP) assay

In direct fluorescence polarization assay, the fluorophore label can interfere with affinity determination and can cause large experimental bias. For this reason, affinities were determined with competitive FP measurements where the bound fluorescent peptide is chased out by increasing concentration of unlabeled competitor. In the fitting procedure, affinities of both peptides are considered. Each PDZ-domain was first tested with labeled peptides in direct experiments and then optimal labeled peptides were used in competitive experiments.

Double affinity-purified His$_6$-MBP-PDZ were used for affinity measurements by FP. Fluorescence polarization was measured with a PHERAstar microplate reader by using $485 \pm 20$ nm and $528 \pm 20$ nm band-pass filters for excitation and emission, respectively. In direct FP measurements, a dilution series of the MBP-PDZ was prepared in 96 well plates (96 well skirted PCR plate, 4ti-0740, 4titude, Wotton, UK) in a 20 mM HEPES pH 7.5 buffer that contained 150 mM NaCl, 0.5 mM TCEP, 0.01% Tween 20 and 50 nM fluorescently-labeled peptide. The volume of the dilution series was 40 μl, which was later divided into three technical replicates of 10 μl during transferring to 384 well micro-plates (low binding microplate, 384 well, E18063G5, Greiner Bio-One, Kremsmünster, Austria). In total, the polarization of the probe was measured at eight different protein concentrations (whereas one contained no protein and corresponded to the free peptide). In competitive FP measurements, the same buffer was supplemented with the protein to achieve a complex formation of 60-80%, based on the direct titration. Then, this mixture was used for preparing a dilution series of the competitor (i.e., the studied peptides) and the measurement was carried out identically as in the direct experiment. Analysis of FP experiments were carried out using ProFit, an in-house developed, Python-based fitting program[70]. The dissociation constants of the direct and competitive FP experiments were obtained by fitting the measured direct data with a quadratic binding equation first and by fitting the measured competitive data with a competitive equation, using several obtained parameters from the first fit[71]. In part, competitive fluorescence polarization data were taken from refs. 22,21. In total, 395 unique interactions were characterized by competitive FP in our database.

## Conversion of holdup binding intensities to dissociation constants

In the holdup measurements, we measure steady-state binding intensities that can be converted to steady-state dissociation constants using Eq. 2:

$$K_d = \frac{([PDZ] - BI^*[PDZ])^*([PBM] - BI^*[PDZ])}{BI^*[PDZ]} \qquad (2)$$

where [PDZ] is the total PDZ concentration (set to 4 μM in usual cases in our assays) and [PBM] is the total peptide concentration. This later parameter is unknown, which makes a direct and accurate conversion impossible. To find the missing peptide concentration, we use orthogonal affinity measurements[21,22]. We determine the dissociation constants by competitive FP of a set of interactions that are also measured by holdup. Then, we substitute the measured BI and $K_d$ values into Eq. (2) and calculate the corresponding [PBM] concentration. Finally, we take the median of all calculated concentrations (3–10 concentrations, depending on peptide) for a given peptide to determine an average parameter that can be used for the conversion of all measured BI values of the same peptide. We have found that in most cases, the peptide concentration lied between 10–40 μM with an average peptide concentration of 18 μM. Only ~39 peptides were characterized by FP, out of the >450 measured by holdup. For peptides that were not characterized by FP, we used this average peptide concentration for conversion.

When we perform the conversion from BI to $K_d$, the binding detection threshold of the holdup measurements dictates the threshold for the quantification of the $K_d$ values in the affinity scale. As a consequence of the peptide concentration and BI detection threshold variability, PDZome-binding profiles of each PBM have a slightly different $K_d$ quantification threshold in affinity scale. In the least sensitive experiment, the $K_d$ quantification threshold was found to be only 30 μM and it was at 790 μM in the most sensitive experiment. On average, the quantification threshold was around $K_d = 320$ μM.

In our database, converted affinities are shown as p$K_d$ values, where p$K_d$ is defined as the negative 10-base logarithm of the dissociation constant. Therefore, p$K_d$ values are closely related to the change in Gibbs free energy upon binding. Due to the different detection thresholds of the different holdup experiments, as well as the different peptide concentrations in each assay, we do not merge together the affinities determined by different holdup methods. Instead, we provide each determined affinities separately for all measured holdup assays, as well as a "composite affinity" for each interaction, which is taken from the most reliable holdup method.

The uncertainty of determined p$K$d values can be estimated in at least two different ways, either using the measured, or propagated uncertainty of BI values, or by measuring the p$K$d variability between the holdup and the FP experiments. Standard deviation of BI values is provided in our database in case more than one experiment was performed. In case of singlicate experiments, one can propagate the errors that we observed in the same kind of experiments when duplicates were measured ($\sigma = 0.05$ for LYSC/SAPF and $\sigma = 0.025$ for DAPF experiments). This standard deviation of BI values can be used to propagate the error into the determined p$K$d value. However, an accurate error propagation model needs to include both the variability of the BI values and the peptide concentration and the latter is difficult to assess based on current data. Alternatively, it is possible to do error estimation based on p$K$d values determined by FP. Note that we use the FP dataset for the determination of the peptide concentration, therefore the two datasets are globally centered. However, each peptide concentration is determined from multiple holdup-FP experiment pairs and therefore the variance of individual pairs can be still measured. The difference between the orthogonal p$K$d values derived from FP and holdup measurements follows a normal distribution with a standard deviation of 0.227 p$K$d. If we assume that neither of these methods perform systematically better than the other we can use this standard deviation as a global metrics of the uncertainty of determined p$K$d values in our database. For example, one can also use it to estimate the confidence of an affinity difference using the following empirical rule: >0.227 p$K$d difference indicates ~68.3% confidence (1 $\sigma$), >0.454 p$K$d difference indicates ~95.5% confidence (2 $\sigma$), and >0.681 p$K$d difference indicates a 99.7% confidence (3 $\sigma$). However, this does not mean that only >0.454 p$K$d differences should be considered as reliable in our database. One needs to assess the uncertainty of each studied measurement in several ways (e.g., by comparing different holdup measurements, or the ways of affinity conversion) in order to make objective judgment on data quality.

## Profaff database

To aid the access to our affinity measurements, we established a PHP- and MySQL-based online database that is accessible at the https://profaff.igbmc.science/ address. On the database, one can browse the affinity profiles of PDZ-domains and PBMs by searching based on gene ID, motif sequence, or consensus motif. It is possible to access all additional information of each interaction, such as measured BI values, details of normalization and conversion, or even the competitive titrations of FP measurements. In addition, for each PDZ-domains where we detected binders above a certain (adjustable) binding

threshold, we provide affinity-weighted frequency plots, based on the sequences of all detected binders, where the weights are calculated based on Eq. (3):

$$weight = 10^{pK_d - binding\,threshold} \qquad (3)$$

Finally, the ProfAff database is built so that it can include any holdup measurements done on other types of domain-motif interaction. For example, in the actual version it also includes measured affinities from some of our previous works: of 14-3-3 proteins with a few phosphopeptides (phosphorylated PBMs)[60] and of HPV E6 proteins with host LxxLL motifs (Bonhoure, A. et al, manuscript in preparation). All these sub-databases are interconnected therefore, it is possible to see whether a PBM is also targeted by 14-3-3 domains, or if a protein also contains an LxxLL site. The code of the database is uploaded to GitHub at https://github.com/GoglG/ProfAff (https://doi.org/10.5281/zenodo.6820648). We hope that the ProfAff database will serve as a proof of concept database for the community and will set a new trend for future interactomic databases.

## Quantitative reference interactome

A non-exhaustive collection of previously published PDZ-PBM affinities used as an external benchmark was generated as follows. We started from the database published by Amacher et al.[29], which makes an inventory of different PDZ-PBM affinities published up to 2007. To complete this list, articles were collected from PubMed using the keywords "PDZ[Title] AND interaction" (April 2021). If an affinity measurement was performed, the nature of the PDZ and the PBM interacting along with their Uniprot codes, the method used for measuring the interaction, the eventual modification to the PBM peptide necessary for the measurement (i.e., fluorophore, biotin, etc), and the measured $K_d$ along with the standard deviation of the measurement (when available), were collected. The same information was collected for the affinities reviewed in Amacher et al. The following data were excluded during collection or from the data reviewed in Amacher et al.: (i) data concerning full-length proteins or PDZ domains in tandem, as opposed to isolated PDZ domains; (ii) data concerning PBM variants (mutations or post-traductional modifications) or artificial PBM sequences (i.e., generated by phage-display); (iii) data for which the length or the sequence of the PBM was not available. The resulting database comprises 1547 $K_d$ measurements, extracted from 132 publications, and measured between 142 different PDZ domains, some of them of different origins (human, mouse, rat, and/or vinegar fly), and 280 different PBM motifs from all classes, including 12 internal PBM. For a complete list of this reference interactome, as well as references for each interaction, see Supplementary Data 1.

For interaction matching we used only affinities from human, mouse, and rat PDZ-domains. PDZ-domain boundaries were not assessed during data collection. In addition, the database contains PBMs with lengths between 4 and >100 residues, but only affinities from 8-12mer peptides were used for interaction matching. Interaction matching was performed based on corresponding 10mer PBM sequences (i.e., exact, extended, or truncated) and PDZ-domain names. In total, we have found 362 matching affinities between our measurements and this filtered benchmark dataset, where the affinities showed a PCC of 0.59 (see Fig. 2B).

## Estimating PPI affinities and matching with a qualitative reference interactome

We used the previously generated domain-motif interactome, consisting of the composite affinities (i.e., taken from the most reliable holdup method; see above) to estimate affinities of PPIs. We assumed that a multidomain PDZ-protein can only interact with a single PBM at the same time (Eq. (4)):

$$PDZ + PBM \rightleftharpoons (PDZ - PBM)_{site1}, \text{or} (PDZ - PBM)_{site2}, \text{or} (PDZ - PBM)_{siten} \qquad (4)$$

Using this approximation, as proposed by others before[45], the association constant of a multi-site interaction is the sum of the association constants of all sites (Eq. (5)):

$$K_{a,additive} = \frac{[PDZ - PBM]_{site1} + [PDZ - PBM]_{site2} + \dots + [PDZ - PBM]_{siten}}{[PDZ] \times [PBM]} = \sum K_{a,site} \qquad (5)$$

This calculation does not imply neither positive, nor negative cooperation between interaction sites. In addition, we do not take into account the formation of higher order, multivalent complexes that may form, especially at higher concentrations. The contribution of unmeasurable weak interactions was also neglected in the calculations. Although for these reasons, our calculation is inherently imperfect, it readily provides a lower limit for the affinity of the "full-length complexes" since most of these neglected effects would in principle increase the global affinity. Most importantly, this simple calculation can be done without any a priori knowledge about the synergistic nature of each multi-domain-proteins.

To collect previously observed qualitatively reported binding events between PDZ and PBM proteins, we analyzed the Biogrid (4.4.199 version), Intact (2020_10_01 version), and Bioplex (Dec_2019 version, both 293T and HCT116 datasets) databases[3,30,31]. We performed the search within the interactomic space that we experimentally studied using the holdup method. To find instances of reported interactions, we looked for co-occurrence of the IDs of PDZ- and PBM-proteins within single entries. In case the PBM originated from a PDZ-protein, this algorithm identifies it as an observed interaction. Thus, any potential self-binding was manually checked in all databases. In total, we have identified 1233, 629, 152, and 56 observed interactions from the Biogrid, Intact, Bioplex (293T) and Bioplex (HCT116) databases. In total, our qualitative benchmark interactome includes 1654 unique interactions observed between the 150 PDZ-proteins and the 448 PBM-proteins. Out of these, 1248 interactions were tested in our assays and we could match quantifiable affinities to 725 interactions.

## Hierarchical clustering of the human PDZ-PBM interactome

To cluster the PDZ-PBM domain-motif interactome based on their interactomic properties, we used unweighted pair-group method with arithmetic average (UPGMA) clustering based on the Euclidean distance in p$K_d$ scale. Note that p$K_d$ values are closely related to binding energies (Eq. 6):

$$pK_d = -\log_{10}K_d = \frac{-\Delta G}{2.303 \times RT} \qquad (6)$$

Note that the Euclidean distance of one affinity in p$K_d$ scale equals to $\Delta\Delta G/2.303*RT$, therefore the Euclidean distance of two binding profile equals to the sum of all $\Delta\Delta G/2.303*RT$ values. Clustering of both PBMs and PDZ domains were performed based on a near-complete part of the interactome consisting of composite affinities, covering 448 PBMs and 133 PDZ-domains. First, we thresholded our interactome at 3.5 p$K_d$, thus removing the very small number of interactions that we managed to quantified at an even weaker threshold. Then, we replaced these and all the unmeasureable weak affinities with 3.5 p$K_d$. Within this interactomic space, the affinities of a few unmeasured interactions ($n < 5$) were estimated based on a kNN (k-Nearest Neighbors) approach. First, a PBM clustering was performed by omitting the affected PDZ-domains. Based on this, the two nearest neighbors in the interactomic space was determined and the missing affinity was replaced by the average of their affinities with the same domain.

## Crystallization

For crystallization, PDZ domains were cloned with an N-terminal TEV protease cleavable His₆ tag and a C-terminal ANXA2 (Annexin A2) tag[72–74]. His₆-PDZ-ANXA2 constructs were produced in E. coli BL21(DE3) with IPTG induction (1 mM IPTG at 18 °C for overnight expression) and harvested cells were lysed in a buffer containing 50 mM TRIS pH 7.5, 150-300 mM NaCl, 2 mM BME, complete EDTA-free protease inhibitor cocktail (Roche, Basel, Switzerland), 1% Triton X-100, and trace amount of DNAse, RNAse, and Lysozyme. Lysates were frozen at −20 °C before further purification steps. Lysates were sonicated and centrifuged for clarification. Expressed PDZ-domains were captured on pre-packed Ni-IDA (Protino Ni-IDA Resin, Macherey-Nagel, Duren, Germany) columns, were washed with at least 10 column volume cold wash buffer (50 mM TRIS pH 7.5, 150 mM NaCl, 2 mM BME) before elution with 250 mM imidazole. The Ni-elution was cleaved with TEV protease and the PDZ-ANXA2 was purified by cation exchange on a HiTrap SP HP column (GE Healthcare, Chicago, Illinois). Proteins were aliquoted at 5–7 mg/ml concentration before flash freezing in LN2. Samples were supplemented with 5–6 molar excess of selected PBM peptides and 2 mM CaCl₂ before crystallization.

Crystallization conditions were screened using commercially available (Qiagen, Hampton Research, Emerald Biosystems) and in-house developed kits by the sitting-drop vapor-diffusion method in 96-well MRC 2-drop plates (SWISSCI, Neuheim, Switzerland), using a Mosquito robot (TTP Labtech, Cambridge, UK). Crystals of MAGI1_2 in complex with the PBM of HPV35 E6 grew rapidly in a drop made from 5 µl of protein solution and 5 µl of reservoir solution containing 20–25% polyethylene glycol 3000, 100 mM sodium citrate buffered at pH 5.5 and 100 mM trisodium-citrate at 20 °C. Crystals of SNTB1 in complex with the PBM of HPV35 E6 grew rapidly in a drop made from 2 µl of protein solution and 2 µl of reservoir solution containing 20% polyethylene glycol 3350, 200 mM sodium malonate buffered at pH 7.0 at 20 °C. Crystals of SYNJ2BP in complex with RPS6KA1_−3P were grown in a drop containing 100 mM ammonium sulfate, 100 mM sodium formate, 25% PEG smear broad (Molecular Dimensions, Sheffield, UK), and 100 mM HEPES pH 7.5 at 20 °C. Crystals of SYNJ2BP in complex with HTLV1 Tax1 were grown in the F10 condition (120 mM Monosaccharides, 100 mM Buffer System 3 pH 8.5, 50% Precipitant Mix 2) of the MORPHEUS screen at 20 °C. Crystals of SNX27 in complex with the PBM of MERS-E grew rapidly in a drop made from 2 µl of protein solution and 2 µl of reservoir solution containing 10% polyethylene glycol 8000, 100 mM imidazole buffered at pH 8.0 at 20 °C. All crystals were flash-cooled in a cryoprotectant solution containing 25% glycerol and stored in liquid nitrogen.

X-ray diffraction data were collected at the Synchrotron Swiss Light Source (Switzerland) on the X06DA (PXIII) beamline or at SOLEIL (France) on the PX2-A beamline. All data were processed with the program XDS[75] and the phase problem was solved by molecular replacement[76], based on the previously determined crystal structures of MAGI1_2-ANXA2 complex (PDB ID 5N7D) and the structures of the PDZ domains of SNTB2, SYNJ2BP, and SNX27 (2VRF, 2JIK, 6SAK) using Phaser and structure refinement was carried out with PHENIX[77]. TLS refinement was applied during the refinement. The crystallographic parameters and the statistics of data collection and refinement are shown in Supplementary Table 1. The refined model and the structure factor amplitudes have been deposited in the PDB with the accession codes 7P70, 7P71, 7P72, 7P73, and 7P74.

## Cell cultures and lysate preparation for proteomic analyses

HEK293T and HeLa cells were authenticated and found 100% identity to ATCC cat. CRL-3216 and ATCC cat. CCL-2, respectively. Jurkat and HaCat cell lines were not authenticated (original source: ECACC and Deutsches Krebsforschungszentrum DKFZ, respectively). HeLa cells were grown in DMEM (1 g/L glucose, Gibco) medium completed with 5% FCS and 40 µg/mL gentamicin, diluted every 3rd/4th day 1/10.

HaCat cells were grown in DMEM (1 g/L glucose, Gibco) medium completed with 10% FCS and 40 µg/mL gentamicin, diluted every 3rd/4th day 1/4. Jurkat cells were grown in RPMI (Gibco) medium completed with 10% FCS and 40 µg/mL gentamicin, diluted every 3rd/4th day 1/12. All cells were kept at 37 °C and 5% CO₂. For proteomics analyses cells were seeded on T-175 flasks. After they reached confluency, adherent cells were washed with PBS once and collected by scraping with ice-cold lysis buffer (Hepes-KOH pH 7.5 50 mM, NaCl 150 mM, Triton X-100 1%, complete EDTA-free protease inhibitor cocktail 1×, EDTA 2 mM, TCEP 5 mM, glycerol 10%). Jurkat cells were collected by 1000 g× centrifugation, washed once with PBS, then collected by 1000 g × 5 min centrifugation again and lysed in ice-cold lysis buffer. Lysates were sonicated 4 × 20 s with 1 s long pulses on ice, then incubated rotating at 4 °C for 30 min. Lysates were centrifuged at 12,000 rpm 4 °C for 20 min and supernatant was kept for further analysis. Concentration determination was carried out by standard Bradford method (Bio-Rad Protein Assay Dye Reagent #5000006) using BSA (MP BIomedicals #160069) as control on a Bio-Rad Smart-Spec 3000 spectrophotometer instrument.

## Sample preparation for affinity purification mass spectrometry (AP-MS) experiments

For PDZ-AP-MS experiments, PDZ domains were cloned as His₆-Avi-Tag-MBP-PDZ constructs. For co-expression with BirA, E. coli BL21(DE3) cells were co-transformed with His₆-AviTag-MBP, or His₆-AviTag-MBP-PDZ-coding and PET21a-BirA (Addgene #20857) plasmids[78]. Simultaneously to IPTG induction, 50 µM biotin was added to the media. After cell harvesting, the lysis buffer was also supplemented with 50 µM biotin. Otherwise, expression and double affinity purification was identical as previously described in details. Biotinylation efficiency was found to be around 60%.

For PBM-AP-MS experiments, 30 µl streptavidin resin was mixed with biotin or peptide at 50–60 µM concentration in 6–6.5 resin volume for 60 min to achieve resin saturation. For PDZ-AP-MS experiments, 30 µl streptavidin resin was mixed with biotinylated MBP or MBP-PDZ at 40-50 µM concentration in >1000× resin volume for 60 min to achieve resin saturation. After saturation, resins were washed a single time (ten resin volume, holdup buffer), and were depleted with biotin (1 mM biotin, 5–10 resin volume, 10 min). Finally, resins were washed two times. 0.5 ml 2 mg/ml cell lysate was added to the dry resin (1 mg total input) and were incubated at 4 °C for 2 h. Then, the beads were washed three times with 10 resin volume buffer containing: 50 mM TRIS pH 8.5, 150 mM NaCl, 1% TritonX-100, 10× complete EDTA-free protease inhibitor cocktail, 2 mM EDTA, 1 mM TCEP. Then, the beads were washed two times with ten resin volume buffer containing: 50 mM TRIS pH 8.5, 150 mM NaCl, 1 mM TCEP. Finally, captured protein was eluted from the resin in two steps and the eluted fractions were pooled. For each elution the beads were incubated for 30 min with three resin volume buffer containing: 20 mM TRIS pH8.5, 100 mM NaCl, 500 µM TCEP, 8 M Urea. Between each step, the beads were separated by mild centrifugation and the supernatant was removed by gentle pipetting.

## Reagents for mass spectrometry

Acetonitrile MS grade, formic acid, trifluoroacetic acid, trichloroacetic acid, iodoacetamide, urea, tris(hydroxymethyl)aminomethane (TRIS), 2-carboxyethy-phosphine from Sigma Aldrich (St. Quentin Fallavier, France), trypsin from Promega (Charbonnieres les Bains, France), lysyl endopeptidase from Wako (Richmond, USA)

## Sample digestion for mass spectrometry

The samples were precipitated with TCA 20% overnight at 4 °C and centrifuged at 14,000 rpm for 10 min at 4 °C. The protein pellets were washed twice with 1 mL cold acetone and air dried. The protein extracts were solubilized in urea 8 M, reduced with 5 mM TCEP for 30 min, and alkylated with 10 mM iodoacetamide for 30 min in the

dark. Double digestion was performed at 37 °C with 500 ng endoproteinase Lys-C for 4 h, followed by fourfold dilution and an overnight digestion with 500 ng trypsin. Peptide mixtures were then desalted on C18 spin-column and dried on Speed-Vacuum.

## LC-MS/MS analysis

Samples were analyzed using an Ultimate 3000 nano-RSLC (Thermo Scientific, San Jose California) coupled in line with a LTQ-Orbitrap ELITE mass spectrometer via a nano-electrospray ionization source (Thermo Scientific, San Jose California). Peptide mixtures were injected in 0.1% TFA on a C18 Acclaim PepMap100 trap-column (75 µm ID × 2 cm, 3 µm, 100 Å, Thermo Fisher Scientific) for 3 min at 5 µL/min with 2% ACN, 0.1% FA in $H_2O$ and then separated on a C18 Accucore nano-column (75 µm ID × 50 cm, 2.6 µm, 150 Å, Thermo Fisher Scientific) at 220 nl/min and 38 °C with a 90 min linear gradient from 5% to 30% buffer B (A: 0.1% FA in $H_2O$/B: 99% ACN, 0.1% FA in $H_2O$), regeneration at 5% B. The mass spectrometer was operated in positive ionization mode, in data-dependent mode with survey scans from $m/z$ 350–1500 acquired in the Orbitrap at a resolution of 120,000 at $m/z$ 400. The 20 most intense peaks from survey scans were selected for further fragmentation in the Linear Ion Trap with an isolation window of 2.0 Da and were fragmented by CID with normalized collision energy of 35%. (TOP20CID method) Unassigned and single charged states were excluded from fragmentation. The Ion Target Value for the survey scans (in the Orbitrap) and the MS2 mode (in the Linear Ion Trap) were set to 1E6 and 5E3 respectively and the maximum injection time was set to 100 ms for both scan modes. Dynamic exclusion was set to 20 s after one repeat count with mass width at ±10 ppm.

## Mass spectrometry data analysis

Proteins were identified by database searching using SequestHT (Thermo Fisher Scientific) with Proteome Discoverer 2.4 software (PD2.4, Thermo Fisher Scientific) on human FASTA database downloaded from SwissProt (reviewed, release 2020_11_27, 20304 entries, https://www.uniprot.org/). Precursor and fragment mass tolerances were set at 7 ppm and 0.6 Da respectively, and up to two missed cleavages were allowed. Oxidation (M, +15.995 Da) was set as variable modification, and Carbamidomethylation (C, +57.021 Da) as fixed modification. Peptides and proteins were filtered with a false discovery rate at 1%. Label-free quantification was based on the extracted ion chromatography intensity of the peptides. All samples were measured in technical triplicates. The measured extracted ion chromatogram (XIC) intensities were normalized based on median intensities of the entire dataset to correct minor loading differences. For statistical tests and enrichment calculations, not detectable intensity values were treated with an imputation method, where the missing values were replaced by random values similar to the 10% of the lowest intensity values present in the entire dataset. Unpaired two-tailed T-test, assuming equal variance, were performed on obtained $\log_2$ XIC intensities. The detection threshold for enriched proteins in AP-MS experiments were analyzed above twofold enrichment with <0.01 $P$ value. Then, proteins with PDZ-domains or PBMs were extracted from the list of enriched proteins. In case only a specific isoform of the detected protein contains a PBM, the identified peptide fragments were manually checked to check which isoform(s) were most likely detected. All raw LC-MS/MS data (from 165 runs) have been deposited to the ProteomeXchange via the PRIDE database with identifier PXD027743.

## Generation of HEK293T stable cell lines

The p10-IRES2-EGFP vector was generated previously[79]. We inserted the wild-type and C-terminal 10 amino acid-deleted (E6ΔPBM) versions of *HPV16 E6* in MCS between the *IR-DRs* by standard restriction enzyme based cloning. Sleeping Beauty transposase containing pSB100 vector

(Addgene #34879) was a kind gift of Dr. Zsuzsanna Izsvak (MDC Berlin Germany) and Dr. Tamas I. Orban (RCNS Budapest Hungary)[80].

HEK293T cells were grown in DMEM (1 g/L glucose, Gibco) medium completed with 10% FCS and 1% penicillin–streptomycin, diluted every 3rd/4th day 1/10, and were kept at 37 °C and 5% $CO_2$. Stable cells were generated as described previously[79]. Briefly, we co-transfected the cells with 2 µg p10 and 0.2 µg pSB100 (10/1 ratio) vectors on six-well plates using jetPRIME (Polyplus #114-15) reagent as recommended by manufacturer. 3 days after transfection cells were trypsinized and sorted for EGFP positive cells on a BD FACS Aria III equipment. Two weeks after the first sort, stable cells were selected by sorting again for EGFP positive cells. Relative copy number of transposon casettes integrated into the genome were determined by qPCR (Supplementary Fig 4C). We do not find significant difference in the copy numbers between E6 and E6ΔPBM expressing cells, however they both contained ~3-times less copy, compared to the control cells. Despite this copy number difference, we do not observe difference in GFP expression in these cells by western blot, which indicates that the expression of GFP from the empty IRES vector is less efficient compared to the vector that also contains E6 or E6ΔPBM before the IRES sequence. Lysates were prepared for proteomics analyses and western blot as described above and below, respectively.

## gDNA isolation and relative copy number determination

gDNA was isolated with standard SDS lysis-isopropanol precipitation method. Relative copy numbers were determined by qPCR with SybrGreen reagent (Quiagen QuantiTect #204143) on a Roche Light-cycler 480 II equipment by using previously described method and primers[81] recognizing the *GFP* and the *IR-DR* left arm (part of the transposon casette) sequence. The Ct values were normalized to *RPPH1* gene using standard ΔΔCt method.

## RT-PCR

Cells were plated on 6-well plates. When they reached confluency, RNA was isolated by using Qiagen RNeasy Mini kit (#74104). On-column DNase digestion was done by using Qiagen RNase-Free DNase Set (#79254). 2 µg total RNA per sample was transcribed to cDNA by using SuperScript IV kit (Invitrogen, #18091050) with random hexamer primers (Thermo Fisher #SO142) according to standard protocol. For qPCR, primers were designed with the help of Primer Blast (NCBI NIH, https://www.ncbi.nlm.nih.gov/tools/primer-blast/) choosing primers which recognize all common alternative transcripts of the given gene (Supplementary Fig 4). qPCR was done by using QuantiTect Syber-Green (Qiagen, #204143) and LightCycler II equipment (Roche). Standard ΔΔCt method was used to determine relative quantity to GAPDH. For further analyses GraphPad Prism 7 software was used.

## Western blot

Cells were directly lysed in 4× Laemmli buffer (120 mM Tris-HCl pH 7, 8% SDS, 100 mM DTT, 32% glycerol, 0.004% bromphenol blue, 1% β-mercaptoethanol). Equal amounts of samples were loaded on 8%, 10%, or 12% acrylamide-gels. Transfer was done into PVDF membranes using a Trans-Blot Turbo Transfer System and Trans-Blot Turbo RTA Transfer Kit (BioRad, #1704273). After 1 h of blocking in 5% milk, membranes were incubated overnight 4 °C in primary antibody in 5% milk. The following antibodies and concentrations were used: anti-GAPDH (Merck #MAB374) 1/5000, anti-HPV16 E6 (IGBMC polyclonal, recognizing the N-terminus of the protein) 1/1000, anti-GFP (IGBMC polyclonal) 1/5000, anti-p53 (CST clone 7F5 #2527) 1/1000, anti-E6AP (Sigma clone 3E5 #SAB1404508-100UG) 1/1000, anti-SCRIB (Thermo Fisher #PA5-54821) 1/1000, anti-MAGI1 (Santa Cruz #sc-100326) 1/1000, anti-SAP97/DLG1 (Thermo Fisher #PA1-741) 1/1000, anti-SNX27 (Thermo Fisher #MA5-27854) 1/500, anti- PDZK1/NHERF3 (Santa Cruz #sc-100337) 1/1000. Membranes were washed three times with TBS-Tween and incubated at RT for 1 h in secondary antibody (Jackson

ImmunoResearch, Peroxidase conjugated Affinipure goat anti-mouse(H + L) #115-035-146 and goat anti-rabbit(H + L) #111-035-003) in 5% milk (concentration 1/10,000). After washing three times with TBS-Tween, membranes were exposed to chemiluminescent HRP substrate (Immobilon, #WBKLS0100) and revealed in docking system (Amersham Imager 600, GE). Densitometry analysis was carried out on raw Tif images by using Fiji ImageJ 1.53c.

### Reporting summary
Further information on research design is available in the Nature Research Reporting Summary linked to this article.

## Data availability
The complete affinity data are provided in Supplementary Data 1 and in our online database ProfAff (https://profaff.igbmc.science), allowing user-friendly visualization and analyses.

The refined models and the structure factor amplitudes of the solved crystal structures have been deposited in the Protein Data Bank (PDB) with the accession codes 7P70, 7P71, 7P72, 7P73, and 7P74. Previously published structures used in this study are available as PDB entries 5N7D, 2VRF, 2JIK, and 6SAK

All raw LC-MS/MS data (from 165 runs) have been deposited to ProteomeXchange via the PRIDE database with identifier PXD027743. Source data are provided with this paper.

## Code availability
The source code for the ProfAff database is available at https://github.com/GoglG/ProfAff and https://doi.org/10.5281/zenodo.6820648.

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

## Acknowledgements

We thank Celia Caillet-Saguy, Nicolas Wolff, and Goran Bich for their help in holdup experiments and data curation, Alastair McEwen and Pierre Poussin-Courmontagne for their help in protein crystallization and data collection, Guillaume Seith for his help in establishing the ProfAff server, Claudine Ebel and Muriel Philipps of flow cytometry platform for their help in cell sorting, Betty Heller for her advises and the cell culture platform for their help in cell culturing, and Ylva Ivvarson for her suggestions with internal motifs. We thank the Swiss Light Source synchrotron (P. Scherrer Institute, Villigen, Switzerland) and the beam-scientists at the PXIII beamline, as well as the PX2A beamline and staff at synchrotron SOLEIL (France). G.G. was supported by the Post-doctorants en France program of the Fondation ARC and B.Z. by Fondation pour la Recherche Médicale (FRM, SPF202005011975). The Travé team was supported by the Ligue contre le cancer (équipe labellisée 2015 to G.T.), the Agence Nationale de la Recherche (grant UBE3A ANR-18-CE92-0017 to G.T.) and the Cancéropôle Grand-Est (projet émergent to Y.N.). As a member of the IGBMC institute, the team also benefited from the French Infrastructure for Integrated Structural Biology (FRISBI) ANR-10-INSB-05-01, from Instruct-ERIC, and from the grant IdEx Unistra (ANR-10-IDEX-0002) provided by SFRI-STRAT'US project (ANR 20-SFRI-0012) and EUR IMCBio (ANR-17-EURE-0023) to the Interdisciplinary Thematic Institute IMCBio, as part of the ITI 2021-2028 program of the University of Strasbourg, CNRS and Inserm under the framework of the French Investments for the Future Program.

## Author contributions

Z.A. and E.M. contributed in experiment design and extensive literature analysis. S.O. and P.E. synthesized the peptides. LYSC type holdup experiments were performed by C.K. and data were analyzed by P.J. and Y.N. SAPF type holdup experiments were performed and analyzed by F.D., R.V., and G.G. DAPF type holdup experiments were performed and analyzed by G.G., B.Z., and C.K. FP measurements were performed by G.G. MS sample preparation and experiments were performed by G.G., B.Z., B.M., and L.N. and were analyzed by B.M. and G.G. Crystallographic experiments were performed by G.G. and A.C. G.G. performed bioinformatics studies, including programing the ProfAff server. C.K. performed all new cloning for this study. B.Z. performed and analyzed all cell-based assays. G.G. and G.T. conceived the idea and supervised the research. G.G., B.Z., E.M., and G.T. wrote the paper. All authors reviewed the manuscript.

## Competing interests

The authors declare no competing interests.
