## [Peer Review File · Nature Communications]

REVIEWER COMMENTS

Reviewer #1 (Remarks to the Author):

Overall, this manuscript represents very nice work from a well-established group pushing the boundaries of our collective understanding of PDZ-binding motifs (PBMs) both in vitro and in the cellular environment. Here, they use extensive proteomics to characterize thousands of PDZ-target interactions, based on known human PDZ domains and their interacting sequences, including those of several viral oncoprotein (e.g., HPV and HTLV). They use fluorescence polarization and crystallographic studies to support their data. These are both well-validated experimental techniques in the PDZ field. There does not appear to be anything too surprising that comes out of their data, but they do establish a very nice experimental pipeline and contribute a plethora of data to the field. Finally, the binding profiles presented and measured correlations between protein concentration and detection in Jurkat/HeLa cells are well appreciated. There are no major criticisms of this paper, but several minor points that the authors should address:

- In general, the manuscript can be edited for flow. There are several instances where it is a bit challenging to understand what the authors are trying to communicate. Some editing for clarity would be useful. For example, the Introduction introduces the idea that the hold-up assay can quantify interactions between $K_d = 100 \mu\text{M}$ to $800 \mu\text{M}$; however, from the language it is a bit unclear if the hold-up assay cannot detect interactions with $K_d < 100 \mu\text{M}$ or just that the majority of interactions are in the $100\text{-}800 \mu\text{M}$ range. It reads as the former ("quantitation thresholds"), which is then nicely shown in Figure 2D as being okay, considering there are likely only ~ 100 interactions in the $K_d < 100 \mu\text{M}$ range. It is disjointed for the reader to have to figure this out for themselves, however. In another location, on pp 7, it states, "to illustrate the unvaluable information provided by our affinity data..." Surely, the authors mean "valuable" here? These are two examples, but the manuscript could be streamlined/edited throughout for clarity.

- The authors state in the Discussion that they "systematically measured 65,000 affinities," however 2/3rd of these interactions were "non-interactions" (the "negatome"), so this statement is a bit misleading. The absence of an interaction, or one weaker than the affinity threshold, is not a "systematic measurement." This should be reworded for clarity and it should be clear that K_d values were quantified for 18,332 interactions. The authors could also make it clear that while PDZ domains are quite promiscuous, binding several targets each and with overlapping specificities to other PDZ domains, it is still quite remarkable that over 18,000 interactions were identified. The authors may choose to reflect on how this ratio (18332/65151) could relate to PDZ-mediated cellular regulation/processes.

- It is unclear how the binding motifs in Figure 3 relate to information from phage display experiments (e.g., Tonikian et al, 2008), and the paper would be stronger if it included such an analysis. Presumably, the phage display data will identify higher affinity interactions, since it is not constrained to endogenous sequences. It would be interesting to know if the binding motifs/biases identified are similar here, particularly considering that several of the same PDZ domains are highlighted.

- The crystal structures from Figure 3 provide nice rationale of the binding motifs/biases, but it is unclear if they were necessary. Do PyMOL alignments of these sequences and previously published structures lead to the same conclusions? Their structures provide a nice proof-of-principle, but presumably as this group explores deeper interactome (or complexome) space, it will be preferable to use computational methods (and known structures) to validate their assays. The Table of data collection and refinement statistics is Table S2 (in the Materials and Methods, it says Table S3 - typo), but it should also be referenced in the main text when the structures are presented.

Reviewer #2 (Remarks to the Author):

Review comments

The authors generated large-scale quantitative interaction data on more than 18,000 PDZ-PDZ binding motif (PDZBM) interaction, and provide also negative data on close to 47,000 interactions. This is an impressive undertaking that will be of great importance for the field. The results are provided in an on-line searchable database, and can also be downloaded from there, which is great. There are some overstatements, but as a whole it is an important contribution to the field.

Minor comments

Page 4. Please provide the number of interactions in the “qualitative benchmark” set. Related to this set, looking into Table S1, there seems to be no references to the primary sources of this set of interaction data (just the databases).

Page 4. The authors suggest that the coverage of the interactome space is sparse. Please specify that this relates to the SLiM based interactome in particular. I don't know if this is true for more stable interactions given the rapid growth of data.

The affinities of most of the interactions measured are of low affinity (100-800 μ M Kd values). Could it be so that most of the “sparsely populated” interaction space is irrelevant (e.g. background noise)? Following this line, to what extent would you expect these interactions to be of functional relevance? It would have been interesting to see some kind of filtering (or enrichment) for interactions that would have higher probability of being functionally relevant based on co-occurrence of PDZBM and PDZ containing proteins in the same cell type, or sharing sub-cellular localization (or shared GO terms). Clearly, the “fragmentomics” approach suffer from the same caveat as many other methods, i.e. potentially mapping interactions between proteins that would not naturally be found in the same cell type (or cell compartment), and it would be helpful for the reader to mention this in the text.

Page 8/9 (and methods). The authors estimated the affinities of multi PDZ proteins for their PDZBM containing targets to be additive. I am not sure how true this is, and it likely varies. The authors could comment on it in the discussion and add some references to detailed studies on the topic.

The last section on proteomic perturbation upon expression of a viral protein is a bit outside of the main scope of the paper. Maybe the authors had different expectations on the outcome, which motivated the addition. It would have been helpful if the authors could have provided a hypothesis that would explain to the reader why this section was included, or somehow make it a more integrated part of the manuscript.

Discussion:

Please use “35,000” rather than 35×10^3 , and 10,000 rather than 10^4 , since you use 65,000 two rows up. The style differences are disturbing the reading.

It would be useful with some kind of indication of what percentage of the 35,000 domains are expected to be peptide binding (as many might have other functions). I am not sure if there is a good estimate of this.

The authors are a bit mixing apples and oranges when comparing the “fragmentomic” estimate of hundreds of millions of potential interactions (which is a likely) with the numbers of interactions reported in the databases and large-scale datasets, which I assume reflect on interactions between proteins. The scales are just different. The authors should make the discussion more balanced to reflect this.

The authors should modify the last sentence of the discussion. It is catchy, but it does imply that papillomavirus is understanding cellular life, which in the extension suggest that the virus has an intelligence..... which seems a bit far stretched for a virus given that it is not even considered to be alive.

Limitations of the study: please remove “yet unlikely” from the last sentence. In the literature there are reported cases where MBP fusions have altered the functions of the fused proteins (e.g. by blocking binding sites). I would argue that it is rather likely that some of the affinities are a bit off, but that’s expected and ok for a large-scale study like this. Finally, the limitation of protein quality is missing. Surely some of the PDZ domains will not be fully functional after freeze thawing. Again, this is ok, but just something to keep in mind for the reader that has little experience of working with purified proteins.

Figures and figure legend

Fig 1. The legend starts with mentioning the scale of the human PDZ-PBM interactome, and then indicate a subsection of it. However, interactions with viral PBMs fall outside of the human interactome, which makes the figure a bit misleading. Clarify this in the legend and in the figure. If you want you could indicate what proportion of the potential host-virus interactome you cover, but that might be a tricky estimate.

Fig 4. The text in this figure is very difficult to read making the figure difficult to understand. You need to organize it in a different way, and could consider moving the comparison with the previously published data to a supplemental figure. The data is complex, so please help the reader.

Thank you for letting me review this manuscript. It was an interesting read.

Reviewer #3 (Remarks to the Author):

This manuscript is focused on understanding the interactions between PDZ domains and their cognate ligands. The authors have taken a high throughput approach to ascertain the binding affinity

(pKd) of the 266 known human PDZ domains and a library of 10 mer peptides comprising 424 human PBMs and an additional 24 viral PBMs.

The authors have extended their previous work using the Hold Up technique. The main technical improvements are changing the assay from a low/medium throughput technique to a high throughput technique that allowed them to perform about 65000 assays to explore approximately 55% of the interaction space. However, the main importance of this work is the depth of coverage in the PDZ::PBM interactions, because much of the technical improvements have been published elsewhere.

The authors have provided enough data to demonstrate that the decrease in accuracy of the high throughput assay is offset by being able to perform more controls and/or conditions and the reported pKd should be viewed with a high degree of confidence.

Although the 55% coverage of the PDZ:PBM space is remarkable, it is unclear why the remaining 45% was not explored. This is a significant weakness of this work, as the power of determining the pKd's is in allowing comparisons between the various PDZ and PBM combinations, as was done in Figure 3. Absent a strong justification for why this space remains unexplored, these missing data points impact the significance of this manuscript, especially if the authors plan to complete exploring the interaction space in the future.

In addition, it would be nice if the authors would try to dig out some more general rules that drive the PDZ::PBM interactions. The examples that are given in this work serve to validate the assay, but do not go much further. For example, the authors have focused on the improvement of binding based on phosphorylation for the interaction between SYNJ2BP and the PBM of RS6KA1, but seem to be ignoring the cases where phosphorylation causes a weakening of binding. With enough data points, the authors might be able to develop some general rules that could be used to predict how the phosphorylation of a PBM effects its binding to a specific PDZ protein - especially for the PBM::PDZ interactions that have weaker/intermediate affinities.

Other points:

The PDZ::PBM ranked plots (Figure 3A, 3D, and Figure 1 upper right panels) seem to hold a lot of information that does not seem to be fully explored. For example, do some exhibit linear decay and others exponential decay? Is there a way to plot this so that all the PDZ domain can be compares, i.e. plotting slope, number of PBMs bound, slope/#PMS, etc? Was Gene Ontology or other analyses used

to explore the binding specificities. For example, is there GO term enrichment overall, just in the best binders, etc.

Figure 2B: There seems to be a significant number of interactions found in the literature with strong affinities that were at or below the limit of detection in the hold up assays. Is there any reason for this?

Figure 3: The authors seem to focus mostly on previously characterized PDZ::PBM interactions and while this is good from a validation standpoint, it makes the work seem less significant as it is difficult to identify any new and interesting insights.

Figure 4 and 5: Is it possible to determine a pKd below which you do not see an interaction by AP-MS? It appears to be around 4.5.

Figure 5: This figure is confusing. A and B left panels are showing very different things, but because they look so similar, it is hard to figure this out. This is made worse by the figure legend which makes it difficult to follow as there are too many lower lefts, and upper rights to keep track of. In addition, there are some comparisons that are being left out: no HaCat plot in 5A left panels and no Jurkat vs Hela in 5B left panels.

Figure 6: The methods for this figure are poorly described. How many replicates were performed, how were the runs quality controlled, etc. Why does it appear that there is a consistent increase of Scrib, Magi, DLG1, and SNX27 in the delPBME6 relative to wtE6 (panel 6E)? Did anything happen to the other viral oncogenes in the 293T cells - E1a and E1b promote the expression and stabilization of p53, and Tag which also binds and stabilizes p53.

We are grateful for all reviewers for their thorough reading and constructive opinions. We modified our manuscript accordingly. In particular, we re-organized the discussion to try to better address and clarify some critical points raised by the reviewers.

The discussion now follows the following steps: i) advantage and perspectives of measuring fragmentomic rather than full-length protein affinities at proteomewide scale, ii) measurability and functional relevance of weak affinities, iii) distinction between "interactomics" and "complexomics", iv) interactomic space and affinity-based interactomic distances, v) application to discover viral interactomic interference hotspots, v) phenotypic impact of an oncoviral PBM as tracked by massive proteomics.

We also realized that the final paragraph "limitations of the present study" did not fit to the format of Nature Communications, therefore we moved the contents of that paragraph to appropriate sections of the material and methods, while taking into account the reviewer comments.

We provide our point-by-point responses below. To help tracking modifications we indicate line counts for the modified parts of the manuscript.

-Responses to REVIEWER COMMENTS

Reviewer #1

#1-1 - In general, the manuscript can be edited for flow. There are several instances where it is a bit challenging to understand what the authors are trying to communicate. Some editing for clarity would be useful.

We edited the text throughout, and re-organized the whole discussion in response to the three reviewers' remarks.

#1-2 For example, the Introduction introduces the idea that the hold-up assay can quantify interactions between $K_d = 100 \mu\text{M}$ to $800 \mu\text{M}$; however, from the language it is a bit unclear if the hold-up assay cannot detect interactions with $K_d < 100 \mu\text{M}$ or just that the majority of interactions are in the $100\text{-}800 \mu\text{M}$ range. It reads as the former ("quantitation thresholds"), which is then nicely shown in Figure 2D as being okay, considering there are likely only ~ 100 interactions in the $K_d < 100 \mu\text{M}$ range. It is disjointed for the reader to have to figure this out for themselves, however.

The limit of quantification of our holdup assay varies a bit from experiment to experiment, and based on our conclusions we specified this range ($K_d = 100 \mu\text{M}$ to $800 \mu\text{M}$) where most of our quantification thresholds were found. Unfortunately, we can not simplify this section entirely, because if we would only report the most likely or average threshold, we would mislead the reader and would have confusing conclusions in later stages. We agree that this part of the manuscript was hard to follow. To clarify the message, we shortened in the result section a lengthy but not essential comment (lines 89-92), which is anyway detailed in the method section (methods lines 1102-1108). The methods also provide key detailed information on the degree of confidence of the data (methods lines 1109-1133).

#1-3 In another location, on pp 7, it states, "to illustrate the invaluable information provided by our affinity data..." Surely, the authors mean "valuable" here? These are two examples, but the manuscript could be streamlined/edited throughout for clarity.

We meant "invaluable" (meaning precious, inestimable, priceless...) This has been corrected (results line 166).

#1-4 - The authors state in the Discussion that they "systematically measured 65,000 affinities," however 2/3rd of these interactions were "non-interactions" (the "negatome"), so this statement is a bit misleading. The absence of an interaction, or one weaker than the affinity threshold, is not a "systematic measurement." This should be reworded for clarity and it should be clear that K_d values were quantified for 18,332 interactions.

-By "measured" we referred here to the act of measuring. We systematically subjected 65,000 distinct PDZ-PBM pairs to individual affinity measurements, and we subsequently obtained quantitative information on the individual affinities of these 65,000 pairs (see below). We edited the first sentence of the discussion to make this clear (line 329).

-Moving from qualitative to quantitative interactomics requires some adjustment of word usage. According to law of mass action, any pair of molecules, even the least attracted to each other, displays a probability of encounter, expressible as a unique, intrinsic affinity constant, which can sample any value in a continuous numerical range. Therefore, the measured 65,000 PDZ-PBM pairs all display affinities. They altogether constitute the "explored interactomic space". In this space, there are no absolute "non-interactions" and no absolute "negatome".

-While all measured pairs display an affinity, only 1/3rd were above the *quantification threshold*, while 2/3rd fell below it. Below such threshold, a traditional qualitative work would consider these pairs as "measured and proven to be non-interactions". As discussed before, these are not non-interactions, but interactions below threshold. The more sensitive the method, the lower the threshold, the higher the proportion of quantified affinities. As a parallel, we can think about the physics of temperature determination. Even today, physicists reach a limit in their capacity to quantify the lowest temperatures. They move closer and closer to "zero Kelvin", but will never reach it in absolute. There is always a lower threshold, where the temperature value cannot be determined, apart from "being below threshold". Yet, any measurement that sets the temperature in the interval between the lower measurable threshold and the absolute zero is still a pretty good quantitative measurement. We will always face the same issue with the weakest affinities.

-We should also remember that, by definition, even the most unfavorable pairs in the PDZ-PBM space display elementary consensus features for mutual recognition. Indeed, we have previously solved crystal structures of unfavorable PDZ/PBM complexes (4JOR.pdb, 6TWX.pdb, 6TWY.pdb) (*Gogl et al Structure 2020*), which were supposedly "non-interactions" "switched off" by phosphorylation or phosphomimicking mutation; and accordingly their affinities fell below holdup assay threshold. Yet these were not structures of "non-interactions": the PBMs were perfectly visible and localized in their classical target groove on the PDZ surface. The affinity constants of these unfavorable pairs could actually be measured by pushing the limits of alternative binding assays, and found to be in the millimolar range not far below the holdup threshold (*Gogl et al Structure 2020*).

These and other comments by the reviewers indicated that we did not sufficiently discuss and clarify these points in our first version of the manuscript. We now include a paragraph dedicated to these issues in the discussion (lines 356-374). We also removed any mention to the "negatome" which was in clear contradiction with our quantitative approach. We also checked throughout the ms that we used terms such as "interactions" or "binding" in their quantitative, relative meaning rather than in the binary, absolute mode.

#1-5 The authors could also make it clear that while PDZ domains are quite promiscuous, binding several targets each and with overlapping specificities to other PDZ domains, it is still quite remarkable that over 18,000 interactions were identified. The authors may choose to reflect on how this ratio (18332/65151) could relate to PDZ-mediated cellular regulation/processes.

-Again, the concepts of "identified" versus "non identified" interactions are not operative in quantitative interactomics. If the holdup assay was sensitive enough, it would in theory have quantified affinities for all the 65,000 PDZ-PBM pairs. The ratio (18332/65151) only relates to the quantification threshold of our assay.

- In a previous work (Jané et al Plos One 2020), we have shown how to calculate a "specificity index" out of PBM- or PDZ-binding profiles obtained by holdup assay. The stronger this index, the more promiscuous the PBM or PDZ considered, relatively to the ensemble of PDZ or PBM targets that have been measured. However, this index is not absolute: it depends on the

interactomic space that has been explored and it can only serve to compare elements belonging to this explored interactomic space. In the present work, we did not calculate such indexes because a large part (95%) of the human PBM-PDZ interactome remains to be measured. Calculation and comparison of promiscuity indexes will only make sense once the full affinity mapping of the PDZ-PBM interactome has been completed.

-As concerns the impact of the PDZ-PBM interactome on cellular regulation and processes, our data mainly confirm previous findings that even the best affinities are in the micromolar range. In the light of cellular abundancies, most of these complexes will only form sporadically in cells. Once all affinities will have been measured, and all protein concentrations will have been quantified in real concentration unit of measurements and in the various cell compartments, the prediction of the concentrations of the many potential, competing or cooperating complexes and of their subsequent functional impact will be certainly an exciting line of future computational research. Currently, we are just starting to develop the affinity measurement approaches that will be needed for collecting affinities at the requested massive scale.

We now added a sentence in the discussion mentioning the promiscuous character of the PDZ-PBM interactome (discussion lines 370-374). We also cited our article by Jané et al 2020 describing the specificity index (results line 160).

#1-6 - It is unclear how the binding motifs in Figure 3 relate to information from phage display experiments (e.g., Tonikian et al, 2008), and the paper would be stronger if it included such an analysis. Presumably, the phage display data will identify higher affinity interactions, since it is not constrained to endogenous sequences. It would be interesting to know if the binding motifs/biases identified are similar here, particularly considering that several of the same PDZ domains are highlighted.

Such a comparison would be interesting in principle, but it would involve serious risks. First, our current PBM peptide library only covers 1/10 of the human PBMome, and is therefore greatly under-sampled as compared to phage display that contains both endogenous and non-natural sequences. Second, phage display profiles are not weighted by affinities in contrast to our profiles. Third, phage display protocols (in particular those of Tonikian et al 2008) include extensive washing procedures that tend to favor interactions with slow dissociation rates (k_{off}). This introduces a frequent bias towards hydrophobic sequences of high affinity yet poor specificity, which are not necessarily representative of sequences naturally occurring in the considered organism. This phenomena was previously analyzed in detail by our group, by focusing precisely on the data of Tonikian et al (*Luck & Travé Bioinformatics 2011*).

#1-7 - The crystal structures from Figure 3 provide nice rationale of the binding motifs/biases, but it is unclear if they were necessary. Do PyMOL alignments of these sequences and previously published structures lead to the same conclusions? Their structures provide a nice proof-of-principle, but presumably as this group explores deeper interactome (or complexome) space, it will be preferable to use computational methods (and known structures) to validate their assays.

-With the structures solved as part of our study, we did not aim to structurally explore the interactome surveyed by affinity measurements, but only to provide original examples on how to analyze the data accessible on the ProfAff database. For example, in the case of the SNX27-bound MERS E (a class 1- type PDZ domain bound to a class 3 peptide), the structure does explain the measured affinity of the interaction that would be difficult to interpret otherwise.

-We fully agree that experimental structural biology cannot realistically reach the pace of quantitative interactomics, whereas cutting edge modeling approaches (i.e. AlphaFold) should be suitable to complement future surveys with systematic structural analysis. However, the debate, whether computational predictions should definitely take over experimental structure solving is not yet closed. Indeed, we recently published a work (Cousido-Siah et al *Acta Cryst D* 2022) where we solved several novel crystal structures of PDZ-PBM complexes and

observed, in some cases, differences between the experimental structures and the alphafold predictions.

In the discussion, we suggest that combining affinity-based fragmentomes and high-throughput structural modeling for proteomewide modeling of protein-protein affinities is a challenging yet exciting avenue for biocomputational research (discussion lines 350-355).

#1-8 The Table of data collection and refinement statistics is Table S2 (in the Materials and Methods, it says Table S3 - typo), but it should also be referenced in the main text when the structures are presented.

The typo is corrected and Table S2 is now cited from the main text (results line 168).

Reviewer #2 (Remarks to the Author):

Review comments

#2-1 The authors generated large-scale quantitative interaction data on more than 18,000 PDZ-PDZ binding motif (PDZBM) interaction, and provide also negative data on close to 47,000 interactions. This is an impressive undertaking that will be of great importance for the field. The results are provided in an on-line searchable database, and can also be downloaded from there, which is great. There are some overstatements, but as a whole it is an important contribution to the field.

Thanks for the positive feedback. In line with the previous responses (#1-4) we would like to mention that we did not provide "negative data" for 47,000 interactions. Instead, we classified all these interactions as "very weak" interactions, which still belong to the PDZ-PBM interactome, and therefore involve *a priori* cognate pairs with an affinity in the millimolar range. This is now clarified in the discussion (discussion lines 360-367) .

Minor comments

#2-1 Page 4. Please provide the number of interactions in the , "qualitative benchmark" set. Related to this set, looking into Table S1, there seems to be no references to the primary sources of this set of interaction data (just the databases).

This qualitative dataset (comprising 1654 interactions) concerns the entire interactomic space between all PDZ and PBM proteins included in our study. However, we only explored a part of this interactome and the covered part of the qualitative dataset includes 1248 described interactions. This was already discussed in details in the method section, but only briefly in the text. We revised the results text to describe our findings more clearly (results lines 116-120).

For the quantitative domain-motif affinity benchmark, we do cite all primary sources. For the qualitative protein-protein benchmark we cite the databases, but cannot cite all primary sources, which are quite numerous. Following our method section, that also provides the exact database versions we used, one can reproduce our finding and easily track back primary sources.

#2-2 Page 4. The authors suggest that the coverage of the interactome space is sparse. Please specify that this relates to the SLiM based interactome in particular. I don't know if this is true for more stable interactions given the rapid growth of data.

As discussed in our responses to #1-4, in quantitative interactomics all proteins display, by principle, a binding affinity. Therefore, the interactomic space includes all potential protein pairs. In human, there are 24,000 proteins, therefore the dimension of the interactomic space is $24,000 \times 24,000 = 576,000,000$ (not including splicing isoforms and all PTM-proteoforms and conformationally distinct forms induced by binding to third parties, which would further increase this number by several orders of magnitude). In Biogrid, the current count of human

protein-protein interactions is around 500,000. Therefore, yes: the current coverage is very sparse for the full-length protein interactome, not only for the sLIM-based interactome. These points are now addressed in more detail in the discussion (discussion lines 330-355). Regardless, in the result section we also added a remark on the fact that our own observation of the sparse coverage is based on SLIM-based interactions (results lines 124-125).

#2-3 The affinities of most of the interactions measured are of low affinity (100-800 μ M Kd values). Could it be so that most of the "sparsely populated" interaction space is irrelevant (e.g. background noise)? Following this line, to what extent would you expect these interactions to be of functional relevance?

-In Fig.2C, we show that the affinities in the explored PDZ-PBM interactome appear to follow an exponential distribution towards the weakest ones. Our pKd quantification threshold, ranging between 4 and 3.1, prevented us from experimentally investigating this trend further. However, as discussed before (#1-4), we only measured *a priori* cognate PDZ-PBM pairs, and we and others have previously crystallized very unfavorable PDZ-PBM complexes whose affinities could still be measured in the millimolar range. These are weak interactions, but they are not "non-interactions", and they are measurable, although not yet at high throughput (but we are currently working on that).

-In our opinion, it would be very risky to decide upon a lower threshold, below which weaker interactions would be considered as "functionally irrelevant". In physics, the relevance of weak nuclear forces is not questioned, because of the existence of strong nuclear forces. In chemistry, nobody questions the relevance of van der Waals forces, because of the existence of covalent bonds. Molecules do not respect binding thresholds but just follow the law of mass action. Their intrinsic affinities and their extrinsic concentrations (which may vary in each cell type and even in each individual cell), define the amount of their complex formation that will eventually contribute to cellular life. For instance, if many weak binders are highly abundant collectively, while the concentrations of the few strongest binders are very low, the summed complexes engaged by the weak binder may end up dominating the scene, in a way that could be determinant for the cellular machinery. These issues have been discussed by previous authors that explicitly pointed to the importance of the numerous weak interactions in biological function (Mc Conkey PNAs 1982, Gierasch and Gershenson Nature Chem Bio 2009). If we want relevant databases, we must quantify, as precisely and completely as possible, all interactions from the strongest to the weakest.

We have clarified these points in the discussion (discussion lines 360-367). We cited in the discussion the above mentioned publications (discussion line 370).

#2-4 It would have been interesting to see some kind of filtering (or enrichment) for interactions that would have higher probability of being functionally relevant based on co-occurrence of PDZBM and PDZ containing proteins in the same cell type, or sharing sub-cellular localization (or shared GO terms). Clearly, the "fragmentomics" approach suffer from the same caveat as many other methods, i.e. potentially mapping interactions between proteins that would not naturally be found in the same cell type (or cell compartment), and it would be helpful for the reader to mention this in the text.

-In the text, we did address this issue, by introducing the general distinction to be made between interactomics and complexomics. Interactomics refer to the intrinsic, context-independent interaction potentials of the system, while complexomics refer to the actual complex formation, as it happens in a context-dependent manner, in different tissues, organelles or situations. In this regard, the reviewer's comment refers to context-dependent complexomics, while our PBM-PDZ pKd dataset pertains to interactomics and is therefore context-independent.

While this distinction between interactomics and complexomics was only hinted in the result section in our prior version, it is now clearly addressed in the discussion (discussion lines 376-385).

-Besides we would like to mention, that the "filtering" should always remain an option that is not applied systematically. Current protein expression data are far from perfect, they are also limited by their own lower quantification thresholds; and we are far from knowing the expression profiles of every cell in complex systems. Any filtering would risk the elimination of key interactions that are only prevalent in special not yet characterized cellular states. Second, as mentioned before, weak interactions may collectively participate in the system, and in this way they may also contribute to the subcellular localization of proteins. So, not accounting for experimentally demonstrated weak-binding information in the modeling or interpretation of cellular processes would be counter-productive. It will be exciting to combine affinity-based interactomes with future proteomics, when we will be able to measure protein concentrations, even at sub-cellular resolution, with absolute concentration unit of measurement. Then, one can use the law of mass action at a system level to generate equilibrium complexomic maps using affinity maps, and these calculations will be only limited to the resolution of absolute proteomics. However, such datasets are not yet available.

#2-5 Page 8/9 (and methods). The authors estimated the affinities of multi PDZ proteins for their PDZBM containing targets to be additive. I am not sure how true this is, and it likely varies. The authors could comment on it in the discussion and add some references to detailed studies on the topic.

The K_a additivity procedure that we employed is a classical "text book approach" (Kitov & Bundle JACS 2003). that we cite in our ms (results line 280, methods line 1190) Yet we agree that many multi-domain proteins might not follow this additivity rule due to positive or negative synergies between domains. In the method section, we raise several potential issues and highlight the limitation of this analysis (methods lines 1193-1200). Yet, we still think it is a useful addition to our study as it allows us to compare fragmentomic results of multidomain proteins with data obtained with full-length proteins.

#2-6 The last section on proteomic perturbation upon expression of a viral protein is a bit outside of the main scope of the paper. Maybe the authors had different expectations on the outcome, which motivated the addition. It would have been helpful if the authors could have provided a hypothesis that would explain to the reader why this section was included, or somehow make it a more integrated part of the manuscript.

There are several reasons why we performed this study.

The first point was to investigate whether the PBM of the oncogenic viral protein E6 had a measurable phenotypic impact on the proteome of E6-expressing cells. The answer was definitely a strong "yes". The data undoubtedly illustrates that the perturbation of particular interaction network will perturb cells at a system level and these perturbation not only concern the direct interaction partners, as often assumed with reductionist logic.

The second point, more anecdotic yet of interest to both HPV and PDZ research fields, was to investigate whether the levels of the strongest E6-binding PDZ-containing proteins were strongly altered by E6 expression. The answer was definitely "no". This is in contrast to many previous reports that E6 should provoke the degradation not only of p53 but also of its PDZ-containing targets. Here, we have confirmed that E6 strongly degrades p53, but in our hands it did not do so for its major PDZ-containing targets. Based on this, one of our conclusive sentences in the discussion is to enlighten the clear difference of mechanism of E6 action on p53 (E6 directly targeting p53 for degradation to wipe it out of the cells) or on PDZ proteins (E6 modulating global PDZ-mediated functions, but with no strong impact on the amount of any individual PDZ protein). We believe that these are interesting and novel findings for our understanding of viral-induced cancers, which add value to the work and reinforce the interest of exhaustively measuring binding affinities .

We edited the corresponding result section (result lines 284-290, 298-300, 306-307) and the discussion (discussion lines 392-398) to better explain the logics beyond these experiments and their relationship to the rest of the study.

Discussion:

#2-7 Please use "35,000" rather than 35×10^3 , and 10,000 rather than 10^4 , since you use 65,000 two rows up. The style differences are disturbing the reading.

This was done throughout the manuscript.

#2-8 It would be useful with some kind of indication of what percentage of the 35,000 domains are expected to be peptide binding (as many might have other functions). I am not sure if there is a good estimate of this.

We are not aware of such estimations, which would be highly useful for future studies. While many small domains are specialized in binding to a particular peptide motif, larger domains (e.g. beta-propeller family members) may display several binding surfaces for several distinct peptide motifs, and many others recognize other types of target molecules.

#2-9 The authors are a bit mixing apples and oranges when comparing the "fragmentomic" estimate of hundreds of millions of potential interactions (which is a likely) with the numbers of interactions reported in the databases and large-scale datasets, which I assume reflect on interactions between proteins. The scales are just different. The authors should make the discussion more balanced to reflect this.

See our response to #2-2. The 24,000 human proteins constitute a binary interactomic space of more than half a billion ($> 500,000,000$) protein pairs, of which only 1/1000th ($\sim 500,000$) are currently qualitatively documented in BioGrid. Furthermore, each individual protein-protein pair can display many alternative binding constants, each associated with highly numerous possible combinations of splicing isoforms, post-translational modifications and conformational changes. So, at the end, the full human interactomic space represents many billions of binding constants to be measured. Yet, interactions of full-length proteins arise from interactions of fragments -domains and motifs, including their potential PTMs. If we boil down to fragmentomics, we can restrain the interactomic space to much smaller sub-interactomes restricted to *a priori* cognate categories (PDZ-PBMs, SH3-polyprolines, SH2-phosphomotifs...), and we eliminate the countless potential combinations cited above. Therefore, boiling down protein-protein interactomes to elementary fragmentomic interactomes certainly decreases the number of affinities to be measured by many orders of magnitude.

Hence the approach suggested in our discussion (discussion lines 330-355): focus on building blocks, measure fragmentomic affinities, then use modeling to integrate them at the more complex level of full-length protein networks. And it will be up to future exciting studies to reveal how fragmentomic affinities relate to affinities of full-length proteins at proteomic scale (discussion lines 352-355). This discussion paragraph also clarifies and resolves the "apple vs pears" distinction between fragmentomic and full-length protein interactomes required by the reviewer.

#2-10 The authors should modify the last sentence of the discussion. It is catchy, but it does imply that papillomavirus is understanding cellular life, which in the extension suggest that the virus has an intelligence, which seems a bit far stretched for a virus given that it is not even considered to be alive.

We took the liberty to make this "joke" in the last conclusive sentence, to illustrate the flaws of reductionism and anthropomorphism in molecular biology. Formulated as it is, unambiguously ironic and at the very last line of the text, it should not harm the scientific rigor of the work. Of course, our aim is not to make readers believe that viruses are intelligent, but instead to lead them to reflect on abusive language frequently used by virologists (including ourselves) to describe "hijacking approaches" or other "smart strategies" "employed" by "viruses to "reprogram" their "hosts" by "targeting" "client proteins" for their "purposes" and so forth... If we can't completely avoid this type of jargon, let us at least remain self-conscious about it. We

modified this last sentence, hoping that it now leaves no ambiguity about our appreciation of viral intelligence (discussion lines 405-406).

#2-11 Limitations of the study: please remove "yet unlikely" from the last sentence. In the literature there are reported cases where MBP fusions have altered the functions of the fused proteins (e.g. by blocking binding sites). I would argue that it is rather likely that some of the affinities are a bit off, but that is expected and ok for a large-scale study like this. Finally, the limitation of protein quality is missing. Surely some of the PDZ domains will not be fully functional after freeze thawing. Again, this is ok, but just something to keep in mind for the reader that has little experience of working with purified proteins.

-Based on all our experience on many test cases over more than twenty years, supported by our past publications dedicated to quality optimization of protein fusions (*Nominé et al Prot Expr Purif 2001, Nominé et al Prot Eng 2001, Zanier et al Prot Expr Purif 2007, Zanier et al JMB 2010, Sidi et al Protein Expr Purif. 2011, Bonhoure et al Microb Cell Fact 2018, Duhoo et al Methods Mol Biol. 2019*) we know that single-domain, easy-folding constructs such as PDZ domains generally do not suffer bias in their binding properties when fused to MBP, while gaining solubility and stability upon storage. In contrast, we have also learnt that tag-free proteins, even subjected to the most careful purification procedure, can often suffer rapid loss of activity, and that this is often counteracted by keeping the protein fused to MBP. In the particular case of our MBP-fused PDZ library, all proteins were quality-checked either with SDS-PAGE or with capillary electrophoresis. Furthermore, the large majority of domains showed detectable binding with at least a few motifs. Yet, it remains possible that some PDZ domains might be particularly unstable and could lose binding activity rapidly during handling. And it is true that a partially inactivated sample may still retain some quantifiable binding, leading to an underestimation of the binding affinity.

Indeed, during revision we realized such a "limitation section" was not compatible with the discussion format of Nature Communications. We therefore relocated this text in appropriate material and methods sections: peptide synthesis for the part concerning biotinylated peptides (methods lines 817-821), and PDZ library for the part concerning MBP-PDZ fusions (methods lines 883-889). We added a cautionary comment to MBP usage accordingly to the reviewer comments (methods lines 887-889). In particular, we removed the "yet unlikely".

Figures and figure legend

#2-12 Fig 1. The legend starts with mentioning the scale of the human PDZ-PBM interactome, and then indicate a subsection of it. However, interactions with viral PBMs fall outside of the human interactome, which makes the figure a bit misleading. Clarify this in the legend and in the figure. If you want you could indicate what proportion of the potential host-virus interactome you cover, but that might be a tricky estimate.

It would be impossible to estimate the number of all viral motifs (including all variants and currently unknown zoonotic viruses, etc). For this reason, we do not include viral motifs on the first panel of the figure and only focused on the human PBMome. On subsequent panels, we clearly separate studied viral and human motifs. We noted the lack of scale for this viral PBMome in the revised figure legend (legend lines 640-641).

#2-13 Fig 4. The text in this figure is very difficult to read making the figure difficult to understand. You need to organize it in a different way, and could consider moving the comparison with the previously published data to a supplemental figure. The data is complex, so please help the reader.

We modified the legend to help the reader (legend lines 696-706). We find it useful to collect here all the obtained AP-MS data on viral motifs not only from our experiments but from other groups which helps to highlight the important proteins involved in the hijacking mechanism. It not only shows the robustness of our assay compared to others, but can also help readers to find the variability of such experiments from lab to lab and from cell line to cell line.

Thank you for letting me review this manuscript. It was an interesting read.

Reviewer #3 (Remarks to the Author):

This manuscript is focused on understanding the interactions between PDZ domains and their cognate ligands. The authors have taken a high throughput approach to ascertain the binding affinity (pKd) of the 266 known human PDZ domains and a library of 10 mer peptides comprising 424 human PBMs and an additional 24 viral PBMs.

The authors have extended their previous work using the Hold Up technique. The main technical improvements are changing the assay from a low/medium throughput technique to a high throughput technique that allowed them to perform about 65000 assays to explore approximately 55% of the interaction space. However, the main importance of this work is the depth of coverage in the PDZ::PBM interactions, because much of the technical improvements have been published elsewhere.

The authors have provided enough data to demonstrate that the decrease in accuracy of the high throughput assay is offset by being able to perform more controls and/or conditions and the reported pKd should be viewed with a high degree of confidence.

#3-1 Although the 55% coverage of the PDZ:PBM space is remarkable, it is unclear why the remaining 45% was not explored. This is a significant weakness of this work, as the power of determining the pKds is in allowing comparisons between the various PDZ and PBM combinations, as was done in Figure 3. Absent a strong justification for why this space remains unexplored, these missing data points impact the significance of this manuscript, especially if the authors plan to complete exploring the interaction space in the future.

-We selected this half of the PDZome (130 PDZ domains) to address interactions of HPV E6, HTLV1 TAX1 and human PBMs of high similarity. This way, we cover nearly all of the interaction partners of these viral motifs with the selected PDZs in our assay, thereby focusing on the "oncoviral hotspot" of the human PDZ-PBM interactome.

We added this explanation to the revised manuscript (results lines 94-97).

-In addition, there was also a technical issue. Our current pipeline requires several "LYSC" standard holdup experiments to determine the partial activities of PDZ domain samples. For the remaining part of the PDZome, we mostly lack this since our previously published holdup measurements were done mostly with HPV E6-like, class I PBMs and therefore our standardisation is mainly valid for PDZs that detectably bind to such motifs. In future developments we aim to eliminate this calibration step, which will allow us to explore the other half of the interactome.

#3-2 In addition, it would be nice if the authors would try to dig out some more general rules that drive the PDZ::PBM interactions. The examples that are given in this work serve to validate the assay, but do not go much further. For example, the authors have focused on the improvement of binding based on phosphorylation for the interaction between SYNJ2BP and the PBM of RS6KA1, but seem to be ignoring the cases where phosphorylation causes a weakening of binding. With enough data points, the authors might be able to develop some general rules that could be used to predict how the phosphorylation of a PBM effects its binding to a specific PDZ protein - especially for the PBM::PDZ interactions that have weaker/intermediate affinities.

-We do systematically provide such specificity determinants within the ProfAff database associated to our paper. For each PDZ domain, one can find affinity weighted LOGOs that highlights general sequence-based rules essential for achieving high affinity. In the manuscript, we have provided several instances of how to exploit such data. However these were just hints - we cannot exhaustively exploit the data in this first article, since this would be lengthy yet still incomplete as we did not fully explore the complete PDZ-PBM interactome.

-Regarding phosphorylation, we do not go in such discussions, since our dataset only contains a handful of phosphorylated PBMs, that were already described elsewhere in detail (Gogl et al, JMB 2019 & Structure 2020). In the absence of interactomic characterization of several new phosphorylated peptides, one can only speculate on general rules based on natural "phosphomimetic" residues. In our previous work (Gogl et al, Structure 2020), we discuss in great details how various position specific phosphorylation will affect PDZ interactions and how the consequences of phosphomimetic substitutions differ from phosphorylation.

-Regarding general rules of PDZ-PBM specificities, we may not be the most competent people to do that. This is typically a challenge for specialized bioinformatic groups, which will hopefully be interested in exploiting our database for such purposes.

Other points:

#3-3 The PDZ::PBM ranked plots (Figure 3A, 3D, and Figure 1 upper right panels) seem to hold a lot of information that does not seem to be fully explored. For example, do some exhibit linear decay and others exponential decay? Is there a way to plot this so that all the PDZ domain can be compares, I.e. plotting slope, number of PBMs bound, slope/#PMS, etc?

-As mentioned before, we cannot fully exploit in this article each direction opened by the very large data delivered. However, as mentioned in the paper, any user of the ProfAff database can easily generate such binding profiles for any domain or motif of one's choice. In addition, pairwise profile comparison is also possible on the website. At the moment it is not possible to perform such calculations on the profiles automatically, but we will consider in the future releases. As for the affinity weighted LOGOS, we do not fully exploit all information in these plots, since such analysis could fill several books.

-As noticed by the reviewer, some highly promiscuous fragments show rather linear binding profile (such as the PDZ domain of SYNJ2BP) while more specific fragments have exponential affinity distributions (such as the MAGI1_2 PDZ domain). Each profile has indeed a different shape, that gives some hints about specificity and promiscuity. One can also use the profiles to generate "promiscuity/specificity indexes" that we have introduced in a former paper (Jané et al Plos One 2021) We have also previously proposed various modes of plotting such data in another work (Jané et al Methods Mol Biol. 2021). Both articles are cited in the revised manuscript (refs 22 and 67). However, the profile shapes and their associated indexes can be biased due to the fact that we have explored only a selected region (6%) of the entire PDZ-PBM interactome, as well as by the detection thresholds which prevent us from accessing to the weakest affinity data, as showed in Fig 2C. We may really be able to quantify promiscuity of human PDZ-PBM interactions once the full PDZ-PBM affinity space will have been measured, and at a lower quantification threshold covering most PDZ-PBM pairs.

#3-4 Was Gene Ontology or other analyses used to explore the binding specificities. For example, is there GO term enrichment overall, just in the best binders, etc.

See our previous answer to #3-2. It cannot be the purpose of this article to explore maximally every potential bioinformatic application of the data. Besides, we attempted GO analysis in several occasions on particular binding profiles, with little success. Generally, we find it difficult to find the most suitable parameters for such analysis since only a part of the proteome contains PDZ domains or PBMs and we only included a small part of these proteins in our assay. However, as we are not experts of this approach, it remains possible that future meta-analysis of our data by expert groups would reveal relevant information.

#3-7 Figure 2B: There seems to be a significant number of interactions found in the literature with strong affinities that were at or below the limit of detection in the hold up assays. Is there any reason for this?

To answer this comment would require to set a lower threshold for "strong affinities in the literature". Let us consider for instance anything better than a Kd of 10 uM (pKd > 5). In this case the figure shows that, out of 362 interactions shown, only 9 "strong literature binders"

correspond to "weak holdup binders" falling in our quantification limit range ($3.5 < PKd \text{ min} < 4$). This is not such a significant number. In our opinion Fig 2B still shows a surprisingly good correlation between our data and the literature, considering all the discrepancies that may occur at many different levels (constructs, methods for purification and storage, methods for binding assays, standards for accuracy and reproducibility... – as mentioned in our manuscript lines 104-106).

#3-8 Figure 3: The authors seem to focus mostly on previously characterized PDZ::PBM interactions and while this is good from a validation standpoint, it makes the work seem less significant as it is difficult to identify any new and interesting insights.

We greatly appreciate the reviewers opinion and we will keep in mind for our follow-up studies. In this work we invested a particular effort to demonstrate the robustness, accuracy and quality of the data. Yet we still believe that we also demonstrated a significant number of novel findings and concepts.

#3-9 Figure 4 and 5: Is it possible to determine a pKd below which you do not see an interaction by AP-MS? It appears to be around 4.5.

We think if someone would like to extrapolate affinities weaker than this apparent threshold using AP-MS, they need to modify our AP-MS pipeline. First, immobilizing more bait (e.g. by changing the resin-lysate ratio) will capture more partners. Second, AP-MS involves an extensive washing protocol which will eliminate weak interactions. By fine-tuning these steps, it should be possible to use the combination of fragmentomics and AP-MS to estimate the affinities of weakly binding full-length proteins from cell lysates.

#3-10 Figure 5: This figure is confusing. A and B left panels are showing very different things, but because they look so similar, it is hard to figure this out. This is made worse by the figure legend which makes it difficult to follow as there are too many lower lefts, and upper rights to keep track of. In addition, there are some comparisons that are being left out: no HaCat plot in 5A left panels and no Jurkat vs HeLa in 5B left panels.

We now greatly simplified the figure legend for Figure 5 (legends lines 717-721). The left panels serve as examples to illustrate the observed correlations between enrichment and fragmentomic affinities, as well as between enrichment measured from two cell extracts. We did not aim to include every protein from every cell line in these plots, but the source data is accessible for the entire dataset in the supplement.

#3-11 Figure 6: The methods for this figure are poorly described. How many replicates were performed, how were the runs quality controlled, etc. Why does it appear that there is a consistent increase of Scrib, Magi, DLG1, and SNX27 in the delPBME6 relative to wtE6 (panel 6E)? Did anything happen to the other viral oncogenes in the 293T cells - E1a and E1b promote the expression and stabilization of p53, and Tag which also binds and stabilizes p53.

We now revised the figure 6 legend (legend lines 734-747) and clarified the corresponding text in the results (results lines 285-307) to make it easier to follow. We did observe a slight increase in the levels of MAGI1 and DLG1 in the presence of the truncated E6. This was found to be significant compared to E6 expressing cells, but not significant compared to the mock cell lines. Also, the E6 expressing cell lines did not show significantly different expression for these proteins compared to the mock cell line. These cell lines were generated using E6, and they do not express other oncogenes of HPV.